# Discovering and deciphering relationships across disparate data modalities

**Joshua T Vogelstein[1,2]\*, Eric W Bridgeford[1], Qing Wang[1], Carey E Priebe[1], Mauro Maggioni[1], Cencheng Shen[3]**

[1]Johns Hopkins University, Baltimore, United States; [2]Child Mind Institute, New York, United States; [3]University of Delaware, Delaware, United States

**Abstract** Understanding the relationships between different properties of data, such as whether a genome or connectome has information about disease status, is increasingly important. While existing approaches can test whether two properties are related, they may require unfeasibly large sample sizes and often are not interpretable. Our approach, 'Multiscale Graph Correlation' (MGC), is a dependence test that juxtaposes disparate data science techniques, including k-nearest neighbors, kernel methods, and multiscale analysis. Other methods may require double or triple the number of samples to achieve the same statistical power as MGC in a benchmark suite including high-dimensional and nonlinear relationships, with dimensionality ranging from 1 to 1000. Moreover, MGC uniquely characterizes the latent geometry underlying the relationship, while maintaining computational efficiency. In real data, including brain imaging and cancer genetics, MGC detects the presence of a dependency and provides guidance for the next experiments to conduct.
DOI: https://doi.org/10.7554/eLife.41690.001

## Introduction

Identifying the existence of a relationship between a pair of properties or modalities is the critical initial step in data science investigations. Only if there is a statistically significant relationship does it make sense to try to decipher the nature of the relationship. Discovering and deciphering relationships is fundamental, for example, in high-throughput screening (*Zhang et al., 1999*), precision medicine (*Prescott, 2013*), machine learning (*Hastie et al., 2001*), and causal analyses (*Pearl, 2000*). One of the first approaches for determining whether two properties are related to—or statistically dependent on—each other is Pearson's Product-Moment Correlation (published in 1895; *Pearson, 1895*). This seminal paper prompted the development of entirely new ways of thinking about and quantifying relationships (see *Reimherr and Nicolae, 2013* and *Josse and Holmes, 2013* for recent reviews and discussion). Modern datasets, however, present challenges for dependence-testing that were not addressed in Pearson's era. First, we now desire methods that can correctly detect any kind of dependence between all kinds of data, including high-dimensional data (such as 'omics), structured data (such as images or networks), with nonlinear relationships (such as oscillators), even with very small sample sizes as is common in modern biomedical science. Second, we desire methods that are interpretable by providing insight into how or why they discovered the presence of a statistically significant relationship. Such insight can be a crucial component of designing the next computational or physical experiment.

While many statistical and machine learning approaches have been developed over the last 120 years to combat aspects of the first issue—detecting dependencies—no approach satisfactorily addressed the challenges across all data types, relationships, and dimensionalities. Hoeffding and Renyi proposed non-parametric tests to address nonlinear but univariate relationships (*Hoeffding, 1948*; *Rényi, 1959*). In the 1970s and 1980s, nearest neighbor style approaches were

**\*For correspondence:**
jovo@jhu.edu

**Competing interests:** The authors declare that no competing interests exist.

**eLife digest** If you want to estimate whether height is related to weight in humans, what would you do? You could measure the height and weight of a large number of people, and then run a statistical test. Such 'independence tests' can be thought of as a screening procedure: if the two properties (height and weight) are not related, then there is no point in proceeding with further analyses.

In the last 100 years different independence tests have been developed. However, classical approaches often fail to accurately discern relationships in the large, complex datasets typical of modern biomedical research. For example, connectomics datasets include tens or hundreds of thousands of connections between neurons that collectively underlie how the brain performs certain tasks. Discovering and deciphering relationships from these data is currently the largest barrier to progress in these fields. Another drawback to currently used methods of independence testing is that they act as a 'black box', giving an answer without making it clear how it was calculated. This can make it difficult for researchers to reproduce their findings – a key part of confirming a scientific discovery. Vogelstein et al. therefore sought to develop a method of performing independence tests on large datasets that can easily be both applied and interpreted by practicing scientists.

The method developed by Vogelstein et al., called Multiscale Graph Correlation (MGC, pronounced 'magic'), combines recent developments in hypothesis testing, machine learning, and data science. The result is that MGC typically requires between one half to one third as big a sample size as previously proposed methods for analyzing large, complex datasets. Moreover, MGC also indicates the nature of the relationship between different properties; for example, whether it is a linear relationship or not.

Testing MGC on real biological data, including a cancer dataset and a human brain imaging dataset, revealed that it is more effective at finding possible relationships than other commonly used independence methods. MGC was also the only method that explained how it found those relationships.

MGC will enable relationships to be found in data across many fields of inquiry – and not only in biology. Scientists, policy analysts, data journalists, and corporate data scientists could all use MGC to learn about the relationships present in their data. To that extent, Vogelstein et al. have made the code open source in MATLAB, R, and Python.

DOI: https://doi.org/10.7554/eLife.41690.002

popularized (*Friedman and Rafsky, 1983*; *Schilling, 1986*), but they were sensitive to algorithm parameters resulting in poor empirical performance. 'Energy statistics', and in particular the distance correlation test (DCORR), was recently shown to be able to detect any dependency with sufficient observations, at arbitrary dimensions, and structured data under a proper distance metric (*Székely et al., 2007*; *Székely and Rizzo, 2009*; *Szekely and Rizzo, 2013*; *Lyons, 2013*). Another set of methods, referred to a 'kernel mean embedding' approaches, including the Hilbert Schmidt Independence Criterion (HSIC) (*Gretton and Gyorfi, 2010*; *Muandet et al., 2017*), have the same theoretical guarantees, which is shown to be a kernel version of the energy statistics (*Sejdinovic et al., 2013*; *Shen and Vogelstein, 2018*). The energy statistics can perform very well with a relatively small sample size on high-dimensional linear data, whereas the kernel methods and another test (Heller, Heller, and Gorfine's test, HHG) (*Heller et al., 2013*) perform well on low-dimensional nonlinear data. But no test performs particularly well on high-dimensional nonlinear data with typical sample sizes, which characterizes a large fraction of real data challenges in the current big data era.

Moreover, to our knowledge, existing dependency tests do not attempt to further characterize the dependency structure. On the other hand, much effort has been devoted to characterizing 'point cloud data', that is, summarizing certain global properties in unsupervised settings (for example, having genomics data, but no disease data). Classic examples of such approaches include Fourier (*Bracewell and Bracewell, 1986*) and wavelet analysis (*Daubechies, 1992*). More recently, topological and geometric data analysis compute properties of graphs, or even higher order simplices (*Edelsbrunner and Harer, 2009*). Such methods build multiscale characterization of the samples,

much like recent developments in harmonic analysis (*Coifman and Maggioni, 2006*; *Allard et al., 2012*). However, these tools typically lack statistical guarantees under noisy observations and are often computationally burdensome.

We surmised that both (i) empirical performance in all dependency structures, in particular high-dimensional, nonlinear, low-sample size settings, and (ii) providing insight into the discovery process, can be addressed via extending existing dependence tests to be *adaptive* to the data (*Zhang et al., 2012*). Existing tests rely on a fixed *a priori* selection of an algorithmic parameter, such as the kernel bandwidth (*Gretton et al., 2006*), intrinsic dimension (*Allard et al., 2012*), and/or local scale (*Friedman and Rafsky, 1983*; *Schilling, 1986*). Indeed, the Achilles Heel of manifold learning has been the requirement to manually choose these parameters (*Levina and Bickel, 2004*). Post-hoc cross-validation is often used to make these methods effectively adaptive, but doing so adds an undesirable computational burden and may weaken or destroy any statistical guarantees. There is therefore a need for statistically valid and computationally efficient adaptive methods.

To illustrate the importance of adapting to different kinds of relationships, consider a simple illustrative example: investigate the relationship between cloud density and grass wetness. If this relationship were approximately linear, the data might look like those in *Figure 1A* (top). On the other hand, if the relationship were nonlinear—such as a spiral—it might look like those in *Figure 1A* (bottom). Although the relationship between clouds and grass is unlikely to be spiral, spiral relationships

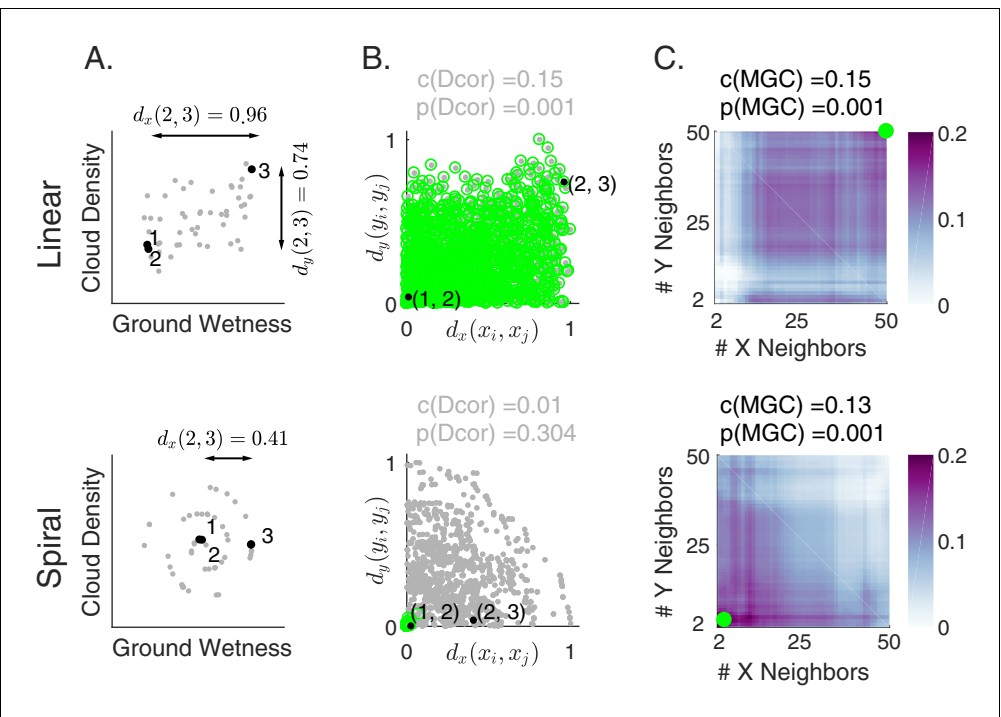

**Figure 1.** Illustration of Multiscale Graph Correlation (MGC) on simulated cloud density ($x_i$) and grass wetness ($y_i$). We present two different relationships: linear (top) and nonlinear spiral (bottom; see Materials and methods for simulation details). (**A**) Scatterplots of the raw data using 50 pairs of samples for each scenario. Samples 1, 2, and 3 (black) are highlighted; arrows show $x$ distances between these pairs of points while their $y$ distances are almost 0. (**B**) Scatterplots of all pairs of distances comparing $x$ and $y$ distances. Distances are linearly correlated in the linear relationship, whereas they are not in the spiral relationship. DCORR uses all distances (gray dots) to compute its test statistic and p-value, whereas MGC chooses the local scale and then uses only the local distances (green dots). (**C**) Heatmaps characterizing the strength of the generalized correlation at all possible scales (ranging from 2 to $n$ for both $x$ and $y$). For the linear relationship, the global scale is optimal, which is the scale that MGC selects and results in a p-value identical to DCORR. For the nonlinear relationship, the optimal scale is local in both $x$ and $y$, so MGC achieves a far larger test statistic, and a correspondingly smaller and significant p-value. Thus, MGC uniquely detects dependence and characterizes the geometry in both relationships.
DOI: https://doi.org/10.7554/eLife.41690.003

are prevalent in nature and mathematics (for example, shells, hurricanes, and galaxies), and are canonical in evaluations of manifold learning techniques (*Lee and Verleysen, 2007*), thereby motivating its use here.

Under the linear relationship (top panels), when a pair of observations are close to each other in cloud density, they also tend to be close to each other in grass wetness (for example, observations 1 and 2 highlighted in black in *Figure 1A*, and distances between them in *Figure 1B*). Similarly, when a pair of observations are far from each other in cloud density, they also tend to be far from each other in grass wetness (see for example, distances between observations 2 and 3). On the other hand, consider the nonlinear (spiral) relationship (bottom panels). Here, when a pair of observations are close to each other in cloud density, they also tend to be close to each other in grass wetness (see points 1 and 2 again). However, the same is not true for large distances (see points 2 and 3). Thus, in the linear relationship, the distance between every pair of points is informative with respect to the relationship, while under the nonlinear relationship, only a subset of the distances are.

For this reason, we juxtapose nearest neighbor mechanism with distance methods. Specifically, for each point, we find its $k$-nearest neighbors for one property (e.g. cloud density), and its $l$-nearest neighbors for the other property (e.g. grass wetness); we call the pair $(k, l)$ the 'scale'. *A priori*, however, we do not know which scales will be most informative. We compute all distance pairs, then efficiently compute the distance correlations for all scales. The local correlations (*Figure 1C*, described in detail below) illustrate which scales are relatively informative about the relationship. The key, therefore, to successfully discover and decipher relationships between disparate data modalities is to adaptively determine which scales are the most informative, and the geometric implication for the most informative scales. Doing so not only provides an estimate of whether the modalities are related, but also provides insight into how the determination was made. This is especially important in high-dimensional data, where simple visualizations do not reveal relationships to the unaided human eye.

Our method, 'Multiscale Graph Correlation' (MGC, pronounced 'magic'), generalized and extends previously proposed pairwise comparison-based approaches by adaptively estimating the informative scales for any relationship — linear or nonlinear, low-dimensional or high-dimensional, unstructured or structured—in a computationally efficient and statistically valid and consistent fashion. This adaptive nature of MGC effectively guarantees an improved statistical performance. Moreover, the dependency strength across all scales is informative about the structure of a statistical relationship, therefore providing further guidance for subsequent experimental or analytical steps. MGC is thus a hypothesis-testing and insight-providing approach that builds on recent developments in manifold and kernel learning, with complementary developments in nearest-neighbor search, and multiscale analyses.

## The multiscale graph correlation procedure

MGC is a multi-step procedure to discover and decipher dependencies across disparate data modalities or properties. Given $n$ samples of two different properties, proceed as follows (see Materials and methods and (*Shen et al., 2018*) for details):

1. Compute two distance matrices, one consisting of distances between all pairs of one property (e.g. cloud densities, entire genomes or connectomes) and the other consisting of distances between all pairs of the other property (e.g. grass wetnesses or disease status). Then center each distance matrix (by subtracting its overall mean, the column-wise mean from each column, and the row-wise mean from each row), and denote the resulting n-by-n matrices $A$ and $B$.

2. For all possible values of $k$ and $l$ from 1 to $n$:
   a. Compute the $k$-nearest neighbor graphs for one property, and the $l$-nearest neighbor graphs for the other property. Let $G_k$ and $H_l$ be the adjacency matrices for the nearest neighbor graphs, so that $G_k(i,j) = 1$ indicates that $A(i,j)$ is within the $k$ smallest values of the $i^{th}$ row of $A$, and similarly for $H_l$.
   b. Estimate the local correlations—the correlation between distances restricted to only the $(k, l)$ neighbors—by summing the products of the above matrices, $c^{kl} = \sum_{ij} A(i,j) G_k(i,j) B(i,j) H_l(i,j)$.
   c. Normalize $c^{kl}$ such that the result is always between $-1$ and $+1$ by dividing by
   $$\sqrt{\sum_{ij} A^2(i,j) G_k(i,j) \times \sum_{ij} B^2(i,j) H_l(i,j)}.$$

3. Estimate the optimal local correlation $c^*$ by finding the smoothed maximum of all local correlations $\{c^{kl}\}$. Smoothing mitigates biases and provides MGC with theoretical guarantee and better finite-sample performance.

4. Determine whether the relationship is significantly dependent—that is, whether $c^*$ is more extreme than expected under the null—via a permutation test. The permutation procedure repeats steps 1–4 on each permutation, thereby eliminating the multiple hypothesis testing problem by only computing one overall p-value, rather than one p-value per scale, ensuring that it is a valid test (meaning that the false positive rate is properly controlled at the specified type I error rate).

Computing all local correlations, the test statistic, and the p-value requires $O(n^2 \log n)$ time, which is about the same running time complexity as other methods (*Shen et al., 2018*).

## Results

### MGC typically requires substantially fewer samples to achieve the same power across all dependencies and dimensions

When, and to what extent, does MGC outperform other approaches, and when does it not? To address this question, we formally pose the following hypothesis test (see Materials and methods for details):

$$H_0 : X \text{ and } Y \text{ are independent}$$
$$H_A : X \text{ and } Y \text{ are not independent.}$$

The standard criterion for evaluating statistical tests is the testing power, which equals the probability that a test correctly rejects the null hypothesis at a given type one error level, that is power = Prob($H_0$ is rejected $|H_0$ is false). The higher the testing power, the better the test procedure. A consistent test has power converging to 1 under dependence, and a valid test controls the type one error level under independence. In a complementary manuscript (*Shen et al., 2018*), we established the theoretical properties of MGC, proving its validity and universal consistency for dependence testing against all distributions of finite second moments.

Here, we address the empirical performance of MGC as compared with multiple popular tests: (i) DCORR, a popular approach from the statistics community (*Székely et al., 2007*; *Székely and Rizzo, 2009*), (ii) MCORR, a modified version of DCORR designed to be unbiased for sample data (*Szekely and Rizzo, 2013*), (iii) HHG, a distance-based test that is very powerful for detecting low-dimensional nonlinear relationships (*Heller et al., 2013*). (iv) HSIC, a kernel dependency measure (*Gretton and Gyorfi, 2010*) formulated in the same way as DCORR except operating on kernels, (v) MANTEL, which is historically widely used in biology and ecology (*Mantel, 1967*). (vi) RV coefficient (*Pearson, 1895*; *Josse and Holmes, 2013*), which is a multivariate generalization of PEARSON's product moment correlation whose test statistic is the sum of the trace-norm of the cross-covariance matrix, and (vii) the CCA method, which is the largest (in magnitude) singular value of the cross-covariance matrix, and can be viewed as a different generalization of PEARSON in high-dimensions that is more appropriate for sparse settings (*Hotelling, 1936*; *Witten et al., 2009*; *Witten and Tibshirani, 2011*). Note that while we focus on high-dimensional settings, Appendix 1 shows further results in one-dimensional settings, also comparing to a number of tests that are limited to one dimension, including: (viii) PEARSON's product moment correlation, (ix) SPEARMAN's rank correlation (*Spearman, 1904*), (x) KENDALL's tau correlation (*Kendall, 1970*), and (xi) MIC (*Reshef et al., 2011*). Under the regularity condition that the data distribution has finite second moment, the first four tests are universally consistent, whereas the other tests are not.

We generate an extensive benchmark suite of 20 relationships, including different polynomial (linear, quadratic, cubic), trigonometric (sinusoidal, circular, ellipsoidal, spiral), geometric (square, diamond, W-shape), and other functions. This suite includes and extends the simulated settings from previous dependence testing work (*Székely et al., 2007*; *Simon and Tibshirani, 2012*; *Gorfine et al., 2012*; *Heller et al., 2013*; *Szekely and Rizzo, 2013*). For many of them, we introduce high-dimensional variants, to more extensively evaluate the methods; function details are in Materials and methods. The visualization of one-dimensional noise-free (black) and noisy (gray) samples is shown in *Figure 2—figure supplement 1*. For each relationship, we compute the power of each

method relative to Mgc for ~20 different dimensionalities, ranging from 1 up to 10, 20, 40, 100, or 1000. The high-dimensional relationships are more challenging because (1) they cannot be easily visualized and (2) each dimension is designed to have less and less signal, so there are many noisy dimensions. *Figure 2* shows that Mgc achieves the highest (or close to the highest) power given 100 samples for each relationship and dimensionality. *Figure 2—figure supplement 2* shows the same advantage in one-dimension with increasing sample size.

Moreover, for each relationship and each method we compute the required sample size to achieve power 85% at error level 0.05, and summarize the median size for monotone relationships (type 1–5) and non-monotone relationships (type 6–19) in *Table 1*. Other methods typically require double or triple the number of samples as Mgc to achieve the same power. More specifically, traditional correlation methods (Pearson, RV, Cca, Spearman, Kendall) always perform the best in monotonic simulations, distance-based methods including Mcorr, Dcorr, Mgc, Hhg and Hsic are slightly worse, while Mic and Mantel are the worst. Mgc's performance is equal to linear methods on monotonic relationships. For non-monotonic relationships, traditional correlations fail to detect the existence of dependencies, Dcorr, Mcorr, and Mic, do reasonably well, but Hhg and Mgc require the fewest samples. In the high-dimensional non-monotonic relationships that motivated this work, and are common in biomedicine, Mgc significantly outperforms other methods. The second best test that is universally consistent (Hhg) requires nearly double as many samples as Mgc, demonstrating that Mgc could half the time and cost of experiments designed to discover relationships at a given effect size.

Mgc extends previously proposed global methods, such as Mantel and Dcorr . The above experiments extended Mcorr , because Mcorr is universally consistent and an unbiased version of Dcorr (*Szekely and Rizzo, 2013*). *Figure 2—figure supplement 3* directly compares multiscale generalizations of Mantel and Mcorr as dimension increases, demonstrating that empirically, Mgc nearly dominates its global variant for essen- tially all dimensions and simulation settings considered here. *Figure 2—figure supplement 4* shows a similar result for one-dimensional settings while varying sample size. Thus, not only does Mgc empirically nearly dominate existing tests, it is a framework that one can apply to future tests to further improve their performance.

## Mgc deciphers latent dependence structure

Beyond simply testing the existence of a relationship, the next goal is often to decipher the nature or structure of the relationship, thereby providing insight and guiding future experiments. A single scalar quantity (such as effect size) is inadequate given the vastness and complexities of possible relationships. Existing methods would require a secondary procedure to characterize the relationship, which introduces complicated 'post selection' statistical quandaries that remain mostly unresolved (*Berk et al., 2013*). Instead, Mgc provides a simple, intuitive, and nonparametric (and therefore infinitely flexible) 'map' of how it discovered the relationship. As described below, this map not only provides interpretability for how Mgc detected a dependence, it also partially characterize the geometry of the investigated relationship.

The Mgc-*Map* shows local correlation as a function of the scales of the two properties. More concretely, it is the matrix of $c^{kl}$'s, as defined above. Thus, the Mgc-Map is an n-by-n matrix which encodes the strength of dependence for each possible scale. *Figure 3* provides the Mgc-Map for all 20 different one-dimensional relationships; the optimal scale to achieve $\hat{t}_*$ is marked with a green dot. For the monotonic dependencies (1-5), the optimal scale is always the largest scale, that is the global one. For all non-monotonic dependencies (6-19), Mgc chooses smaller scales. Thus, a global optimal scale implies a close-to-linear dependency, otherwise the dependency is strongly nonlinear. In fact, this empirical observation led to the following theorem (which is proved in Materials and methods):

*Theorem 1. When $(X, Y)$ are linearly related (meaning that $Y$ can be constructed from $X$ by rotation, scaling, translation, and/or reflection), the optimal scale of Mgc equals the global scale. Conversely, a local optimal scale implies a nonlinear relationship.*

Thus, the Mgc-Map explains how Mgc discovers relationships, specifically, which scale has the most informative pairwise comparisons, and how that relates to the geometry of the relationship. Note that Mgc provides the geometric characterization 'for free', meaning that no separate procedure is required; therefore, Mgc provides both a valid test and information about the geometric relationship.

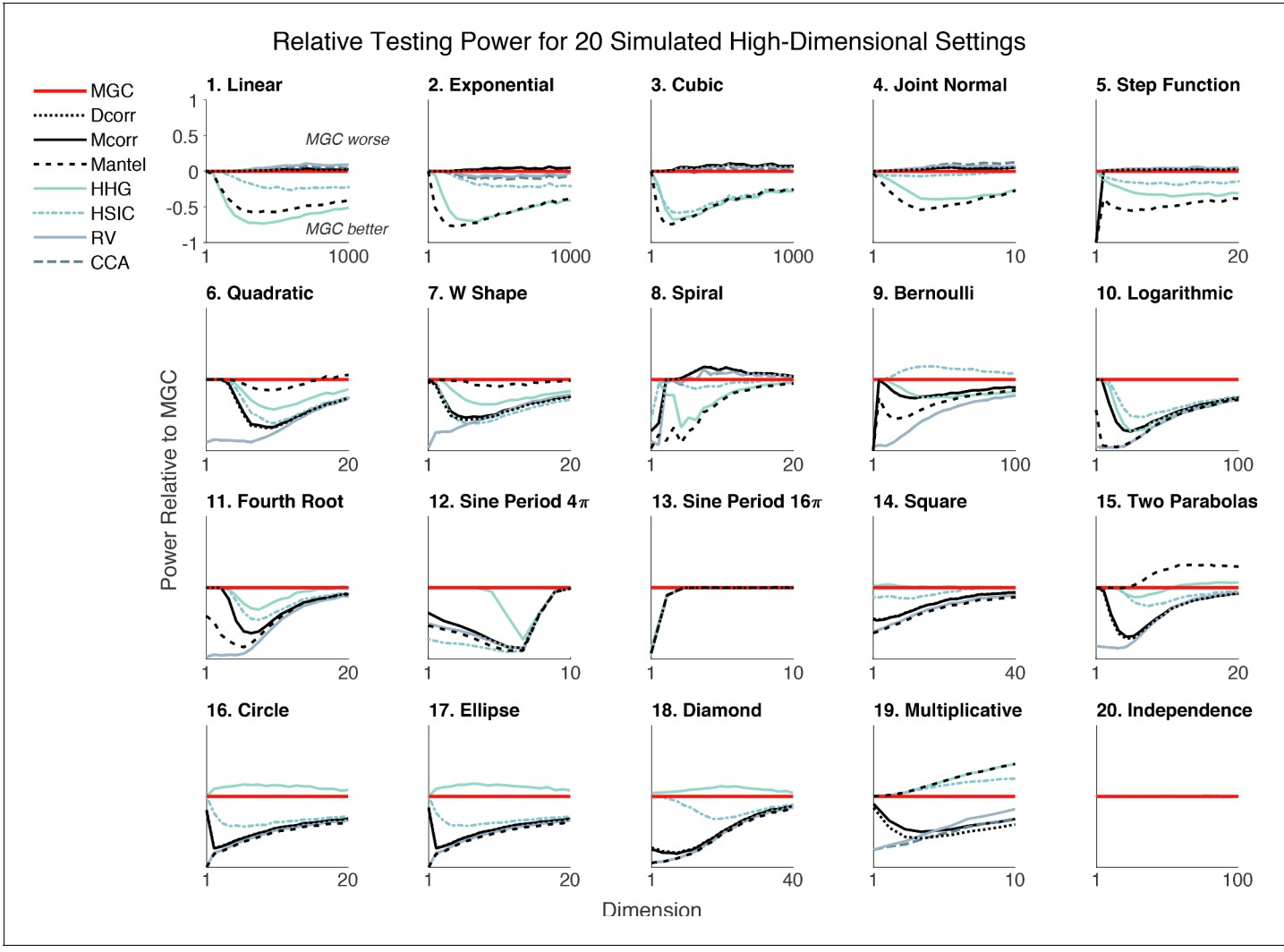

**Figure 2.** An extensive benchmark suite of 20 different relationships spanning polynomial, trigonometric, geometric, and other relationships demonstrates that MGC empirically nearly dominates eight other methods across dependencies and dimensionalities ranging from 1 to 1000 (see Materials and methods and *Figure 2—figure supplement 1* for details). Each panel shows the testing power of other methods relative to the power of MGC (e.g. power of MCORR minus the power of MGC) at significance level $\alpha = 0.05$ versus dimensionality for $n = 100$. Any line below zero at any point indicates that that method's power is less than MGC's power for the specified setting and dimensionality. MGC achieves empirically better (or similar) power than all other methods in almost all relationships and all dimensions. For the independent relationship (#20), all methods yield power 0.05 as they should. Note that MGC is always plotted 'on top' of the other methods, therefore, some lines are obscured.
DOI: https://doi.org/10.7554/eLife.41690.004

The following figure supplements are available for figure 2:

**Figure supplement 1.** Visualization of the 20 dependencies at $p = q = 1$.
DOI: https://doi.org/10.7554/eLife.41690.005
**Figure supplement 2.** The same power plots as in *Figure 2*, except the 20 dependencies are one-dimensional with noise, and the x-axis shows sample size increasing from 5 to 100.
DOI: https://doi.org/10.7554/eLife.41690.006
**Figure supplement 3.** The same set-ups as in *Figure 2*, comparing different MGC implementations versus its global counterparts.
DOI: https://doi.org/10.7554/eLife.41690.007
**Figure supplement 4.** The same power plots as in *Figure 3*, except the 20 dependencies are one-dimensional with noise, and the x-axis shows sample size increasing from 5 to 100.
DOI: https://doi.org/10.7554/eLife.41690.008

**Table 1.** The median sample size for each method to achieve power 85% at type one error level 0.05, grouped into monotone (type 1–5) and non-monotone relationships (type 6–19) for both one- and ten-dimensional settings, normalized by the number of samples required by MGC.

In other words, a 2.0 indicates that the method requires double the sample size to achieve 85% power relative to MGC. PEARSON, RV, and CCA all achieve the same performance, as do SPEARMAN and KENDALL. MGC requires the fewest number of samples in all settings, and for high-dimensional non-monotonic relationships, all other methods require about double or triple the number of samples MGC requires.

| Dimensionality | One-Dimensional | | | Ten-Dimensional | | |
| --- | --- | --- | --- | --- | --- | --- |
| Dependency type | Monotone | Non-Mono | Average | Monotone | Non-Mono | Average |
| MGC | 1 | 1 | 1 | 1 | 1 | 1 |
| DCORR | 1 | 2.6 | 2.2 | 1 | 3.2 | 2.6 |
| MCORR | 1 | 2.8 | 2.4 | 1 | 3.1 | 2.6 |
| HHG | 1.4 | 1 | 1.1 | 1.7 | 1.9 | 1.8 |
| HSIC | 1.4 | 1.1 | 1.2 | 1.7 | 2.4 | 2.2 |
| MANTEL | 1.4 | 1.8 | 1.7 | 3 | 1.6 | 1.9 |
| PEARSON / RV / CCA | 1 | >10 | >10 | 0.8 | >10 | >10 |
| SPEARMAN / KENDALL | 1 | >10 | >10 | n/a | n/a | n/a |
| MIC | 2.4 | 2 | 2.1 | n/a | n/a | n/a |

DOI: https://doi.org/10.7554/eLife.41690.009

The following source data is available for Table 1:
**Source data 1.** Testing power sample size data in one dimension.
DOI: https://doi.org/10.7554/eLife.41690.010
**Source data 2.** Testing power sample size data in high-dimensions.
DOI: https://doi.org/10.7554/eLife.41690.011

Moreover, similar dependencies have similar MGC-Maps and often similar optimal scales. For example, logarithmic (10) and fourth root (11), although very different functions analytically, are geometrically similar, and yield very similar MGC-Maps. Similarly, (12) and (13) are trigonometric functions, and they share a narrow range of significant local scales. Both circle (16) and ellipse (17), as well as square (14) and diamond (18), are closely related geometrically and also have similar MGC-Maps. This indicates that the MGC-Map partially characterizes the geometry of these relationships, differentiating different dependence structures and assisting subsequent analysis steps. Moreover, in *Shen and Vogelstein, 2018*, we proved that the sample MGC-Map (which MGC estimates) converges to the true MGC-Map provided by the underlying joint distribution of the data. In other words, each relationship has a specific map that characterizes it based on its joint distribution, and MGC is able to accurately estimate it via sample observations. The existence of a population level characterization of the joint distribution strongly differentiates MGC from previously proposed multi-scale geometric or topological characterizations of data, such as persistence diagrams (*Edelsbrunner and Harer, 2009*).

## MGC is computationally efficient

MGC does not incur large computational costs and has a similar complexity as existing methods. Though a naïve implementation of MGC requires $\mathcal{O}(n^4)$ operations, we devised a nested implementation that requires only $\mathcal{O}(n^2 \log n)$ operations. Moreover, obtaining the MGC-Map costs no additional computation, whereas other methods would require running a secondary computational step to decipher geometric properties of the relationship. MGC can also trivially be parallelized, reducing computation to $\mathcal{O}(n^2 \log n/T)$, where $T$ is the number of cores (see Algorithm C1 for details). Since $T$ is often larger than $\log n$, in practice, MGC can be $\mathcal{O}(n^2)$, meaning only a constant factor slower than DCORR and HSIC, which is illustrated in Figure 6 of *Shen and Vogelstein, 2018*. For example, at sample size $n = 5000$ and dimension $p = 1$, on a typical laptop computer, DCORR requires around 0.5 s to compute the test statistic, whereas MGC requires no more than 5 s. But the cost and time to obtain 2.5× more data (so DCORR has same average power as MGC) typically far exceeds a few seconds. In

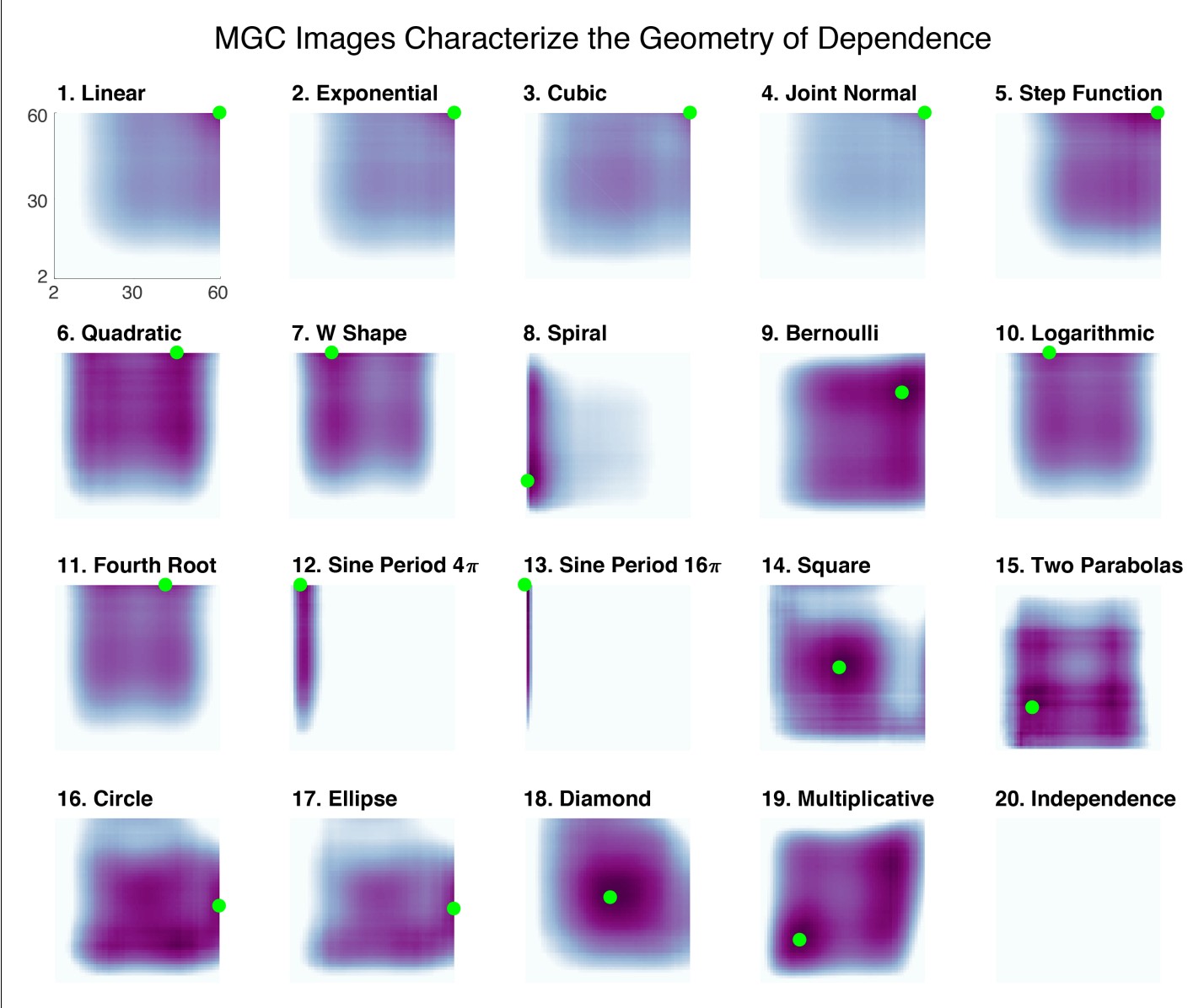

**Figure 3.** The MGC-Map characterizes the geometry of the dependence function. For each of the 20 panels, the abscissa and ordinate denote the number of neighbors for $X$ and $Y$, respectively, and the color denotes the magnitude of each local correlation. For each simulation, the sample size is 60, and both $X$ and $Y$ are one-dimensional. Each dependency has a different MGC-Map characterizing the geometry of dependence, and the optimal scale is shown in green. In linear or close-to-linear relationships (first row), the optimal scale is global, that is the green dot is in the top right corner. Otherwise the optimal scale is non-global, which holds for the remaining dependencies. Moreover, similar dependencies often share similar MGC-Maps and similar optimal scales, such as (10) logarithmic and (11) fourth root, the trigonometric functions in (12) and (13 , 16) circle and (17) ellipse, and (14) square and (18) diamond. The MGC-Maps for high-dimensional simulations are provided in *Figure 3—figure supplement 1*.

DOI: https://doi.org/10.7554/eLife.41690.012

The following figure supplement is available for figure 3:

**Figure supplement 1.** The MGC-Map for the 20 panels for high-dimensional dependencies.

DOI: https://doi.org/10.7554/eLife.41690.013

comparison, the cost to compute a persistence diagram is typically $\mathcal{O}(n^3)$, which is orders of magnitude slower when $n > 10$. The running time of each method on the real data experiments are reported in Materials and methods.

## Mgc uniquely reveals relationships in real data

Geometric intuition, numerical simulations, and theory all provide evidence that Mgc will be useful for real data discoveries. Nonetheless, real data applications provide another necessary ingredient to justify its use in practice. Below, we describe several real data applications where we have used Mgc to understand relationships in data that other methods were unable to provide.

### Mgc discovers the relationships between brain and mental properties

The human psyche is of course dependent on brain activity and structure. Previous work has studied two particular aspects of our psyche: personality and creativity, developing quantitative metrics for evaluating them using structured interviews (*Costa and McCrae, 1992*; *Jung et al., 2009*). However, the relationship between brain activity and structure, and these aspects of our psyche, remains unclear (*DeYoung et al., 2010*; *Xu and Potenza, 2012*; *Bjørnebekk et al., 2013*; *Sampaio et al., 2014*). For example, prior work did not evaluate the relationship between entire brain connectivity and all five factors of the standard personality model (*Costa and McCrae, 1992*). We therefore utilized Mgc to investigate published open access data (see Materials and methods for details).

First, we analyzed the relationship between resting-state functional magnetic resonance (rs-fMRI) activity and personality (*Adelstein et al., 2011*). The first row of *Table 2* compares the p-value of different methods, and *Figure 4A* shows the Mgc-Map for the sample data. Mgc is able to yield a significant p-value (< 0.05), whereas all previously proposed global dependence tests under consideration (Mantel, Dcorr, Mcorr, or Hhg) fail to detect dependence at a significance level of 0.05. Moreover, the Mgc-Map provides a characterization of the dependence, for which the optimal scale indicates that the dependency is strongly nonlinear. Interestingly, the Mgc-Map does not look like any of the 20 images from the simulated data, suggesting that the nonlinearity characterizing this dependency is more complex or otherwise different from those we have considered so far.

Second, we investigated the relationship between diffusion MRI derived connectivity and creativity (*Jung et al., 2009*). The second row of *Table 2* shows that Mgc is able to ascertain a dependency between the whole brain network and the subject's creativity. The Mgc-Map in *Figure 4B* closely resembles a linear relationship where the optimal scale is global. The close-to-linear relationship is also supported from the p-value table as all methods except Hsic are able to detect significant dependency, which suggests that there is relatively little to gain by pursuing nonlinear regression techniques, potentially saving valuable research time by avoiding tackling an unnecessary problem. The test statistic for both Mgc and Mcorr equal 0.04, which is quite close to zero despite a significant p-value, implying a relatively weak and noisy relationship. A prediction of creativity via linear regression turns out to be non-significant, which implies that the sample size is too low to obtain useful predictive accuracy (not shown), indicating that more data are required for single subject predictions. If one had first directly estimated the regression function, obtaining a null result, it would remain unclear whether a relationship existed. This experiment demonstrates that for high-dimensional and potentially structured data, Mgc is able to reveal dependency with relatively small sample size while parametric techniques and directly estimating regression functions can often be ineffective.

**Table 2.** The p-values for brain imaging vs mental properties.
Mgc *always* uncovers the existence of significant relationships and discovers the underlying optimal scales. Bold indicates significant p-value per dataset.

| Testing Pairs/Methods | Mgc | Dcorr | Mcorr | Hhg | Hsic |
|---|---|---|---|---|---|
| Activity vs Personality | **0.043** | 0.667 | 0.441 | 0.059 | 0.124 |
| Connectivity vs Creativity | **0.011** | **0.010** | **0.011** | **0.031** | 0.092 |

DOI: https://doi.org/10.7554/eLife.41690.015

The following source data is available for Table 2:
**Source data 1.** p-value data for activity vs personality.
DOI: https://doi.org/10.7554/eLife.41690.016

**Source data 2.** p-value data for connetivity vs creativity.
DOI: https://doi.org/10.7554/eLife.41690.017

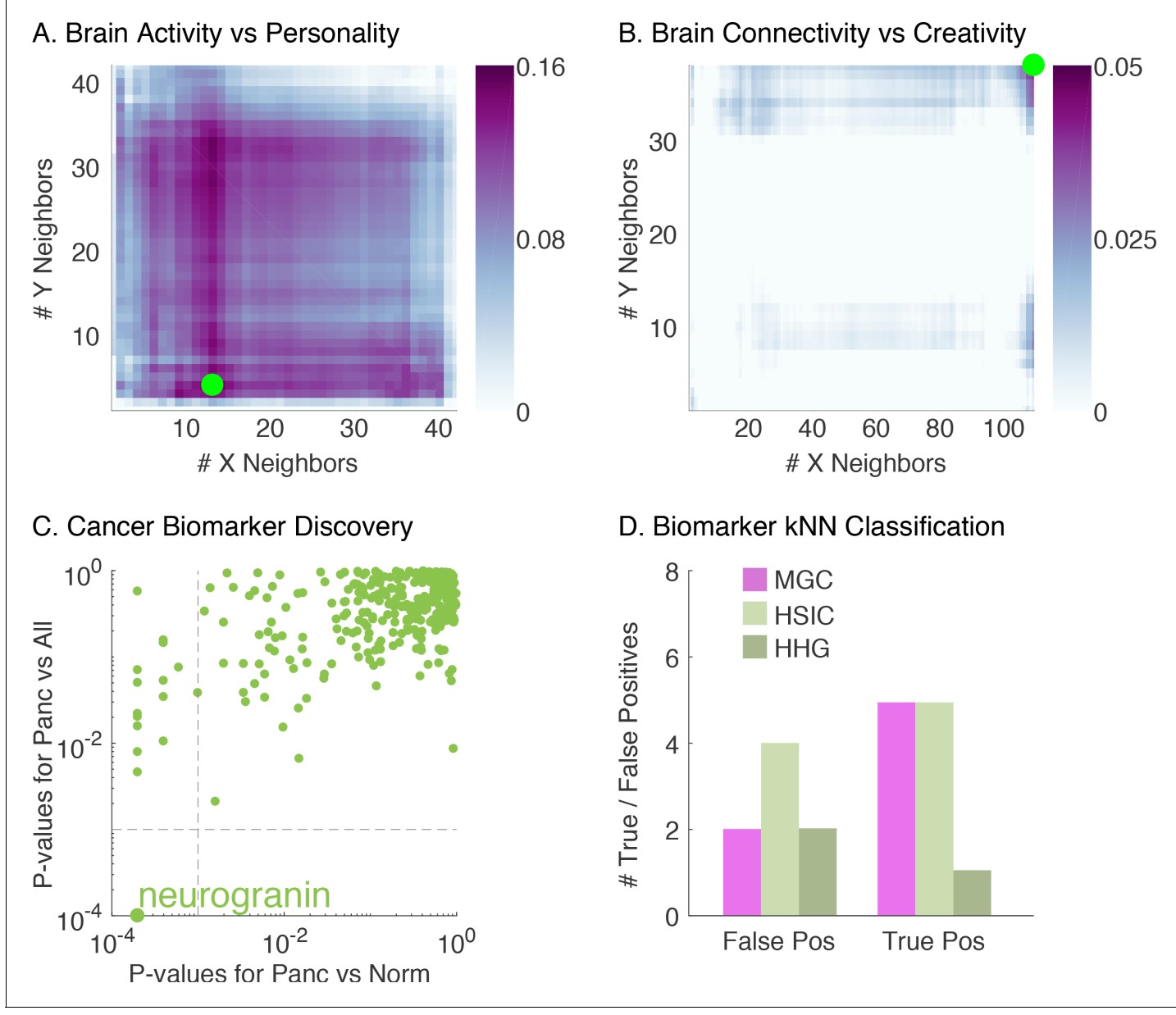

**Figure 4.** Demonstration that MGC successfully detects dependency, distinguishes linearity from nonlinearity, and identifies the most informative feature in a variety of real data experiments. (**A**) The MGC-Map for brain activity versus personality. MGC has a large test statistic and a significant p-value at the optimal scale (13, 4), while the global counterpart is non-significant. That the optimal scale is non-global implies a strongly nonlinear relationship. (**B**) The MGC-Map for brain connectivity versus creativity. The image is similar to that of a linear relationship, and the optimal scale equals the global scale, thus both MGC and MCORR are significant in this case. (**C**) For each peptide, the x-axis shows the p-value for testing dependence between pancreatic and healthy subjects by MGC, and the y-axis shows the p-value for testing dependence between pancreatic and all other subjects by MGC. At critical level 0.05, MGC identifies a unique protein after multiple testing adjustment. (**D**) The true and false positive counts using a k-nearest neighbor (choosing the best $k \in [1, 10]$) leave-one-out classification using only the significant peptides identified by each testing method. The peptide identified by MGC achieves the best true and false positive rates, as compared to the peptides identified by HSIC or HHG.
DOI: https://doi.org/10.7554/eLife.41690.014

The performance in the real data closely matches the simulations results: the first dataset exhibits a strongly nonlinear relationship, for which MGC has the lowest p-value, followed by HHG and HSIC and then all other methods; the second dataset exhibits a close-to-linear relationship, for which global methods perform the best while HHG and HSIC are trailing. Moreover, MGC detected a complex nonlinear relationship for brain activity versus personality, and a nearly linear but noisy

relationship for brain network versus creativity, the only method able to make either of those claims. In a separate experiment, we assessed the frequency with which MGC obtained false positive results using brain activity data, based on experiments from *Eklund et al. (2012)*; *Eklund et al. (2016)*. *Appendix 1—figure 1* shows that MGC achieves a false positive rate of 5% when using a significance level of 0.05, implying that it correctly controls for false positives, unlike typical parametric methods on these data.

## MGC identifies potential cancer proteomics biomarkers

MGC can also be useful for a completely complementary set of scientific questions: screening proteomics data for biomarkers, often involving the analysis of tens of thousands of proteins, peptides, or transcripts in multiple samples representing a variety of disease types. Determining whether there is a relationship between one or more of these markers and a particular disease state can be challenging but is a necessary first step (*Frantzi et al., 2014*). We sought to discover new useful protein biomarkers from a quantitative proteomics technique that measures protein and peptide abundance called Selected Reaction Monitoring (SRM) (*Wang et al., 2011*). Specifically, we were interested in finding biomarkers that were unique to pancreatic cancer, because it is lethal and no clinically useful biomarkers are currently available (*Bhat et al., 2012*).

The data consist of proteolytic peptides derived from the blood samples of 95 individuals harboring pancreatic ($n = 10$), ovarian ($n = 24$), colorectal cancer ($n = 28$), and healthy controls ($n = 33$). The processed data included 318 peptides derived from 121 proteins. Previously, we used these data and other techniques to find ovarian cancer biomarkers (a much easier task because the dataset has twice as many ovarian patients) and validated them with subsequent experiments (*Wang et al., 2017*). Therefore, our first step was to check whether MGC could correctly identify ovarian biomarkers. Indeed, the peptides that have been validated previously are also identified by MGC. Emboldened, using the same dataset, we applied MGC to screen for biomarkers unique to pancreatic cancer. To do so, we first screened for a difference between pancreatic cancer and healthy controls, identifying several potential biomarkers. Then, we screened for a difference between pancreatic cancer and all other conditions, to find peptides that differentiate pancreatic cancer from other cancers. *Figure 4C* shows the p-value of each peptide assigned by MGC, which reveals one particular protein, neurogranin, that exhibits a strong dependency specifically with pancreatic cancer. Subsequent literature searches reveal that neurogranin is a potentially valuable biomarker for pancreatic cancer because it is exclusively expressed in brain tissue among normal tissues and has not been linked with any other cancer type (*Yang et al., 2015*; *Willemse et al., 2018*). In comparison, HSIC identified neurogranin as well, but it also identified another peptide; HHG identified the same two by HSIC, and a third peptide. A literature evaluation of these additional peptides shows that they are upregulated in other cancers as well and are unlikely to be useful as a pancreatic biomarker (*Helfman et al., 2018*; *Lam et al., 2012*). The rest of the global methods did not identify any markers at significance level 0.05, see Materials and methods for more details and *Appendix 1—table 2* for identified peptide information using each method.

Since there is no ground truth yet in this experiment, we further carried out a classification task using the biomarkers identified by the various algorithms, using a k-nearest-neighbor classifier to predict pancreatic cancer, and a leave-one-subject-out validation. *Figure 4D* shows that the peptide selected by MGC (neurogranin) works better than any other subset of the peptides selected by HSIC or HHG, in terms of both fewer false positives and negatives. This analysis suggests MGC can effectively be used for screening and subsequent classification.

## Discussion

There are a number of connections between MGC and other prominent statistical procedures that may be worth further exploration. First, MGC can be thought of as a regularized or sparsified variant of distance or kernel methods. Regularization is central to high-dimensional and ill-posed problems, where dimensionality is larger than sample size. Second, MGC can also be thought of as learning a metric because it chooses the optimal scale amongst a set of $n^2$ truncated distances, motivating studying the relationship between MGC and recent advances in metric learning (*Xing et al., 2003*). In particular, deep learning can be thought of as metric learning (*Giryes et al., 2015*), and generative adversarial networks (*Goodfellow et al., 2014*) are implicitly testing for equality, which is closely

related to dependence (*Sutherland et al., 2016*). While Mgc searches over a two-dimensional parameter space to optimize the metric, deep learning searches over a much larger parameter space, sometimes including millions of dimensions. Probably neither is optimal, and somewhere between the two would be useful in many tasks. Third, energy statistics provide state of the art approaches to other problems, including goodness-of-fit (*Székely and Rizzo, 2005*), analysis of variance (*Rizzo and Székely, 2010*), conditional dependence (*Székely and Rizzo, 2014*; *Wang et al., 2015*), and feature selection (*Li et al., 2012*; *Zhong and Zhu, 2015*), so Mgc can be adapted for them as well. Indeed, Mgc can also implement a two-sample (or generally the K-sample) test (*Szekely and Rizzo, 2004*; *Heller et al., 2016*; *Shen and Vogelstein, 2018*). Specifically, for more than two modalities, one may use summation of pairwise Mgc test statistics, similar to how energy statistic is generalized to K-sample testing from two-sample testing (*Rizzo and Székely, 2010*; *Rizzo and Székely, 2016*; *Shen and Vogelstein, 2018*), or how canonical correlation analysis is generalized into more than two modalities (*Kettenring, 1971*; *Tenenhaus and Tenenhaus, 2011*; *Shen et al., 2014*). Finally, although energy statistics have not yet been explicitly used for classification, regression, or dimensionality reduction, Mgc opens the door to these applications by providing guidance as to how to proceed. Specifically, it is well documented in machine learning literature that the choice of kernel, metric, or scale often has a strong effect on the performance of different machine learning algorithms (*Levina and Bickel, 2004*). Mgc provides a mechanism to estimate scale that is both theoretically justified and computationally efficient, by optimizing a metric for a task wherein the previous methods lacked a notion of optimization. Nonlinear dimensionality reduction procedures, such as Isomap (*Tenenbaum et al., 2000*) and local linear embedding (*Roweis and Saul, 2000*) for example, must also choose a scale, but have no principled criteria for doing so. Mgc could be used to provide insight into multimodal dimensionality reduction as well.

The default metric choice of Mgc in this paper is always the Euclidean distance, but other metric choices may be more appropriate in different fields, and using the strong negative type metric as specified in *Lyons (2013)* guarantees consistency. However, if multiple metric choices are experimented to yield multiple Mgc p-values, then the optimal p-value should be properly corrected for multiple testing. Alternatively, one may use the maximum Mgc statistic among multiple metric choices, apply the same procedure in each permutation (i.e. in each permutation, use the same number of metric choices and take the maximum Mgc as the permuted statistic), then derive a single p-value. Such a testing procedure properly controls the type one error level without the need for additional correction.

Mgc also addresses a particularly vexing statistical problem that arises from the fact that methods methods for discovering dependencies are typically dissociated from methods for deciphering them. This dissociation creates a problem because the statistical assumptions underlying the 'deciphering' methods become compromised in the process of 'discoverying'; this is called the 'post-selection inference' problem (*Berk et al., 2013*). The most straightforward way to address this issue is to collect new data, which is costly and time-consuming. Therefore, researchers often ignore this fact and make statistically invalid claims. Mgc circumvents this dilemma by carefully constructing its permutation test to estimate the scale in the process of estimating a p-value, rather than after. To our knowledge, Mgc is the first dependence test to take a step towards valid post-selection inference.

As a separate next theoretical extension, we could reduce the computational space and time required by Mgc. Mgc currently requires space and time that are quadratic with respect to the number of samples, which can be costly for very large data. Recent advances in related work demonstrated that one could reduce computational time of distance-based tests to close to linear via faster implementation, subsampling, random projection, and null distribution approximation (*Huo and Székely, 2016*; *Huang and Huo, 2017*; *Zhang et al., 2018*; *Chaudhuri and Hu, 2018*), making it feasible for large amount of data. Alternately, semi-external memory implementations would allow running Mgc even as the interpoint comparison matrix exceeds the size of main memory (*Da Zheng et al., 2015*; *Da Zheng et al., 2016a*; *Da Zheng et al., 2016b*; *Da Zheng et al., 2016c*).

Finally, Mgc is easy to use. Source code is available in MATLAB, R, and Python from https://mgc. neurodata.io/ (*Bridgeford et al., 2018*; experiments archived at https://github.com/elifesciences-publications/MGC-paper). Code for reproducing all the figures in this manuscript is also available from the above websites. We showed Mgc's value in diverse applications spanning neuroscience (which motivated this work) and an 'omics example. Applications in other domains facing similar

questions of dependence, such as finance, pharmaceuticals, commerce, and security, could likewise benefit from MGC.

## Materials and methods

### Mathematical details

This section contains essential mathematical details on independence testing, the notion of the generalized correlation coefficient and the distance-based correlation measure, how to compute the local correlations, and the smoothing technique. A statistical treatment on MGC is in *Shen and Vogelstein, 2018*, which introduces the population version of MGC and various theoretical properties.

#### Testing independence

Given pairs of observations $(x_i, y_i) \in \mathbb{R}^p \times \mathbb{R}^q$ for $i = 1, \ldots, n$, assume they are independently identically distributed as $(X, Y) \overset{iid}{\sim} F_{XY}$. If the two random variables $X$ and $Y$ are independent, the joint distribution equals the product of the marginals, that is $F_{XY} = F_X F_Y$. The statistical hypotheses for testing independence is as follows:

$$H_0 : F_{XY} = F_X F_Y,$$

$$H_A : F_{XY} \neq F_X F_Y.$$

Given a test statistic, the testing power equals the probability of rejecting the independence hypothesis (i.e. the null hypothesis) when it is false. A test statistic is consistent if and only if the testing power increases to 1 as sample size increases to infinity. We would like a test to be universally consistent, that is consistent against all joint distributions. DCORR, MCORR, HSIC, and HHG are all consistent against any joint distribution of finite second moments and finite dimension.

Note that $p$ is the dimension for $x$'s, $q$ is the dimension for $y$'s. For MGC and all benchmark methods, there is no restriction on the dimensions, that is the dimensions can be arbitrarily large, and $p$ is not required to equal $q$. The ability to handle data of arbitrary dimension is crucial for modern big data. There also exist some special methods that only operate on one-dimensional data, such as (*Reshef et al., 2011*; *Heller et al., 2016*; *Huo and Székely, 2016*), which are not directly applicable to multidimensional data.

#### Correlation measures

To achieve consistent testing, most state-of-the-art dependence measures operate on pairwise comparisons, either similarities (such as kernels) or dissimilarities (such as distances).

Let $\mathcal{X}_n = \{x_1, \cdots, x_n\} \in \mathbb{R}^{p \times n}$ and $\mathcal{Y}_n = \{y_1, \cdots, y_n\} \in \mathbb{R}^{q \times n}$ denote the matrices of sample observations, and $\delta_x$ be the distance function for $x$'s and $\delta_y$ for $y$'s. One can then compute two $n \times n$ distance matrices $\tilde{A} = \{\tilde{a}_{ij}\}$ and $\tilde{B} = \{\tilde{b}_{ij}\}$, where $\tilde{a}_{ij} = \delta_x(x_i, x_j)$ and $\tilde{b}_{ij} = \delta_y(y_i, y_j)$. A common example of the distance function is the Euclidean metric ($L^2$ norm), which serves as the starting point for all methods in this manuscript.

Let $A$ and $B$ be the transformed (e.g., centered) versions of the distance matrices $\tilde{A}$ and $\tilde{B}$, respectively. Any 'generalized correlation coefficient' (*Spearman, 1904*; *Kendall, 1970*) can be written as:

$$c(\mathcal{X}_n, \mathcal{Y}_n) = \frac{1}{z} \sum_{i=1}^{n} \sum_{j=1}^{n} a_{ij} b_{ij}, \tag{1}$$

where $z$ is proportional to the standard deviations of $A$ and $B$, that is $z = n^2 \sigma_a \sigma_b$. In words, $c$ is the global sample correlation across *pairwise comparison matrices* $A$ and $B$, and is normalized into the range $[-1, 1]$, which usually has expectation 0 under independence and implies a stronger dependency when the correlation is further away from 0.

Traditional correlations such as the Pearson's correlation and the rank correlation can be written via the above correlation formulation, by using $A$ and $B$ directly from sample observations rather than distances. Distance-based methods like DCORR and MANTEL operate on the Euclidean distance

by default, or other metric choices on the basis of domain knowledge; then transform the resulting distance matrices $\tilde{A}$ and $\tilde{B}$ by certain centering schemes into $A$ and $B$. Hsic chooses the Gaussian kernel and computes two kernel matrices, then transform the kernel matrices $\tilde{A}$ and $\tilde{B}$ by the same centering scheme as Dcorr. For Mgc, $A$ and $B$ are always distance matrices (or can be transformed to distances from kernels by *Sejdinovic et al. (2013)*), and we shall apply a slightly different centering scheme that turns out to equal Dcorr.

To carry out the hypothesis testing on sample data via a nonparametric test statistic, for example a generalized correlation, the permutation test is often an effective choice (*Good, 2005*), because a p-value can be computed by comparing the correlation of the sample data to the correlation of the permuted sample data. The independence hypothesis is rejected if the p-value is lower than a predetermined type 1 error level, say 0.05. Then the power of the test statistic equals the probability of a correct rejection at a specific type 1 error level. Note that Hhg is the only exception that cannot be cast as a generalized correlation coefficient, but the permutation testing is similarly effective for the Hhg test statistic; also note that the *iid* assumption is critical for permutation test to be valid, which may not be applicable in special cases like auto-correlated time series (*Guillot and Rousset, 2013*).

## Distance correlation (Dcorr) and the Unbiased Version (Mcorr)

Define the row and column means of $\tilde{A}$ by $\bar{a}_{\cdot j} = \frac{1}{n}\sum_{i=1}^{n}\tilde{a}_{ij}$ and $\bar{a}_{i\cdot} = \frac{1}{n}\sum_{j=1}^{n}\tilde{a}_{ij}$. Dcorr defines

$$a_{ij} = \begin{cases} \tilde{a}_{ij} - \bar{a}_{i\cdot} - \bar{a}_{\cdot j} + \bar{a}, & \text{if } i \neq j, \\ 0, & \text{if } i = j, \end{cases}$$

and similarly for $b_{ij}$. For distance correlation, the numerator of *Equation 1* is named the distance covariance (Dcov), while $\sigma_a$ and $\sigma_b$ in the denominator are the square root of each distance variance. The centering scheme is important to guarantee the universal consistency of Dcorr, whereas Mantel uses a simple centering scheme and thus not universally consistent.

Let $c(X, Y)$ be the population distance correlation, that is, the distance correlation between the underlying random variables $X$ and $Y$. *Székely et al. (2007)* define the population distance correlation via the characteristic functions of $F_X$ and $F_Y$, and show that the population distance correlation equals zero if and only if $X$ and $Y$ are independent, for any joint distribution $F_{XY}$ of finite second moments and finite dimensionality. They also show that as $n \to \infty$, the sample distance correlation converges to the population distance correlation, that is, $c(\mathcal{X}_n, \mathcal{Y}_n) \to c(X, Y)$. Thus the sample distance correlation is consistent against any dependency of finite second moments and dimensionality. Of note, the distance covariance, distance variance, and distance correlation are always non-negative. Moreover, the consistency result holds for a much larger family of metrics, those of strong negative type (*Lyons, 2013*).

It turns out that the sample distance correlation has a finite-sample bias, especially as the dimension $p$ or $q$ increases (*Szekely and Rizzo, 2013*). For example, for independent Gaussian distributions, the sample distance correlation converges to 1 as $p, q \to \infty$. By excluding the diagonal entries and slightly modifies the off-diagonal entries of $\mathcal{A}$ and $\mathcal{B}$, Szekely and Rizzo (*Szekely and Rizzo, 2013*; *Székely and Rizzo, 2014*) show that Mcorr is an unbiased estimator of the population distance correlation $c(\boldsymbol{x}, \boldsymbol{y})$ for all $p, q, n$, which is approximately normal even if $p, q \to \infty$. Thus it enjoys the same theoretical consistency as Dcorr and always has zero mean under independence.

## Local correlations

Given any matrices $A$ and $B$, we can define a set of local correlations as follows. Let $R(A_{\cdot j}, i)$ be the 'rank' of $\boldsymbol{x}_i$ relative to $\boldsymbol{x}_j$, that is, $R(A_{\cdot j}, i) = k$ if $\boldsymbol{x}_i$ is the $k^{th}$ closest point (or 'neighbor') to $\boldsymbol{x}_j$, as determined by ranking the $n - 1$ distances to $x_j$. Define $R(B_{i\cdot}, j)$ equivalently for the $Y$'s, but ranking relative to the rows rather than the columns (see below for explanation). For any neighborhood size $k$ around each $\boldsymbol{x}_i$ and any neighborhood size $l$ around each $\boldsymbol{y}_j$, we define the local pairwise comparisons:

$$\widetilde{a}_{ij}^k = \begin{cases} a_{ij}, & \text{if } R(A_{\cdot j}, i) \leq k, \\ 0, & \text{otherwise}; \end{cases} \qquad \widetilde{b}_{ij}^l = \begin{cases} b_{ij}, & \text{if } R(B_{i\cdot}, j) \leq l, \\ 0, & \text{otherwise}; \end{cases} \qquad (2)$$

and then let $a_{ij}^k = \widetilde{a}_{ij}^k - \bar{a}^k$, where $\bar{a}^k$ is the mean of $\{\widetilde{a}_{ij}^k\}$, and similarly for $b_{ij}^l$.

The *local* correlation coefficient at a given scale is defined to effectively exclude large distances:

$$c^{kl}(\mathcal{X}_n, \mathcal{Y}_n) = \frac{1}{z_{kl}} \sum_{i,j=1}^{n} a_{ij}^k b_{ij}^l, \tag{3}$$

where $z_{kl} = n^2 \sigma_a^k \sigma_b^l$, with $\sigma_a^k$ and $\sigma_b^l$ is the standard deviations for the truncated pairwise comparisons. The MGC-Map can be constructed by computing all local correlations, which allows the discovery of the optimal correlation. For any aforementioned correlation (DCORR, MCORR, HSIC, MANTEL, PEARSON), one can define its local correlations by using *Equation 3* and plugging in the respective $a_{ij}$ and $b_{ij}$ from *Equation 1*.

As most nonlinear relationships intrinsically exhibit a local linear structure, considering the nearest-neighbors is able to amplify the dependency signal over the global correlation. There could be two other scenarios: when the small distances in one modality mostly correspond to large distances in another modality, or when the large distances in one modality correspond to large distance in another modality. For the first scenario, the small distances become negative terms after centering while the large distances become positive terms after centering, so adding their product to $c^{kl}$ will cause the test statistic to be smaller — in fact, as distance correlation is shown to be > 0 under dependence (*Székely et al., 2007*), the first scenario cannot happen for all distances pairs. For the second scenario, one can experiment using the large distances (or the furthest neighbors) only by reversing the ranking scheme in local correlation to descending order. However, whenever the large distances are highly correlated, the small distances must also be highly correlated after centering by the mean distances, so global correlation coefficient like DCORR already handles this scenario. Therefore considering the nearest-neighbor may significantly improve the performance over global correlation, while considering the other scenarios does not.

## MGC as the optimal local correlation

We define the multiscale graph correlation statistic as the optimal local correlation, for which the family of local correlation is computed based on Euclidean distance and MCORR transformation.

Instead of taking a direct maximum, MGC takes a smoothed maximum, that is the maximum local correlation of the largest connected component $R$ such that all local correlations within $R$ are significant. If no such region exists, MGC defaults the test statistic to the global correlation (details in Algorithm C2). Thus, we can write:

$$c^*(\mathcal{X}_n, \mathcal{Y}_n) = \max_{(k,l) \in R} c^{kl}(\mathcal{X}_n, \mathcal{Y}_n)$$

$$R = \text{Largest Connected Component of } \{(k,l) \text{ such that } c^{kl} > \max(\tau, c^{nn})\}.$$

Then the optimal scale equals all scales within $R$ whose local correlations are as large as $c^*$. The choice of $\tau$ is made explicit in the pseudo-code, with further discussion and justification offered in *Shen and Vogelstein, 2018*.

## Proof for theorem 1

**Theorem 1**. When $(X, Y)$ are linearly related (rotation, scaling, translation, reflection), the optimal scale of MGC equals the global scale. Conversely, that. the optimal scale is local implies a nonlinear relationship.

*Proof.* It suffices to prove the first statement, then the second statement follows by contrapositive. When $(X, Y)$ are linearly related, $Y = WX + b$ for a unitary matrix $W$ and a constant $b$ up-to possible scaling, in which case the distances are preserved, that is $\|y_i - y_j\| = \|Wx_i - Wx_j\| = \|x_i - x_j\|$. It follows that $\text{MCORR}(\mathcal{X}_n, \mathcal{Y}_n) = 1$, so the global scale achieves the maximum possible correlation, and the largest connected region $R$ is empty. Thus the optimal scale is global and $\text{MGC}(\mathcal{X}_n, \mathcal{Y}_n) = \text{MCORR}(\mathcal{X}_n, \mathcal{Y}_n) = 1$.

## Computational complexity of each step

The distance computation takes $\mathcal{O}(n^2 \max\{p, q\})$, and the ranking process takes $\mathcal{O}(n^2 \log n)$. Once the distance and ranking are completed, computing one local generalized correlation requires $\mathcal{O}(n^2)$

(see Algorithm C4). Thus, a naive approach to compute all local generalized correlations requires at least $\mathcal{O}(n^2 \max\{n^2, p, q\})$ by going through all possible scales, meaning possibly $\mathcal{O}(n^4)$ which would be computationally prohibitive. However, given the distance and ranking information, we devised an algorithm that iteratively computes all local correlations in $\mathcal{O}(n^2)$ by re-using adjacent smaller local generalized correlations (see Algorithm C5). Therefore, when including the distance computation and ranking overheads, the MGC statistic is computed in $\mathcal{O}(n^2 \max\{\log n, p, q\})$, which has the same running time as the HHG statistic, and the same running time up to a factor of $\log n$ as global correlations like DCORR and MCORR, which require $\mathcal{O}(n^2 \max\{p, q\})$ time. By utilizing a multi-core architecture, MGC can be computed in $\mathcal{O}(n^2 \max\{\log n, p, q\}/T)$ instead. As $T = \log(n)$ is often a small number, for example $T$ is no more than 30 at 1 billion samples, thus MGC can be effectively computed in the same complexity as DCORR. Note that the permutation test adds another $r$ random permutations to the $n^2$ term, so computing the p-value requires $\mathcal{O}(n^2 \max\{\log n, p, q, r\}/T)$.

## MGC algorithms and testing procedures

Six algorithms are presented in order:

- Algorithm C1 describes MGC in its entirety (which calls most of the other algorithms as functions).
- Algorithm C2 computes the MGC test statistic.
- Algorithm C3 computes the p-value of MGC by the permutation test.
- Algorithm C4 computes the local generalized correlation coefficient at a given scale $(k, l)$, for a given choice of the global correlation coefficient.
- Algorithm C5 efficiently computes all local generalized correlations, in nearly the same running time complexity as computing one local generalized correlation.
- Algorithm C6 evaluates the testing power of MGC by a given distribution.

For ease of presentation, we assume there are no repeating observations of $X$ or $Y$, and note that MCORR is the global correlation choice that MGC builds on.

---

**Pseudocode C1** Multiscale Graph Correlation (MGC); requires $\mathcal{O}(n^2 \max(\log n, p, q, r)/T)$ time, where $r$ is the number of permutations and $T$ is the number of cores available for parallelization.

---

**Input:** $n$ samples of $(x_i, y_i)$ pairs, an integer $r$ for the number of random permutations.

---

**Output:** (i) MGC statistic $c^*$, (ii) the optimal scale $(k, l)$, (iii) the p-value $p(c^*)$,

---

 **function** MG($(x_i, y_i)$, for $i \in [n]$)

 **(1)** Calculate all pairwise distances:

 **for** $i, j := 1, \ldots, n$ **do**

 $a_{ij} = \delta_x(x_i, x_j)$ $\delta_x$ is the distance between pairs of $x$ samples

 $b_{ij} = \delta_y(y_i, y_j)$ $\delta_y$ is the distance between pairs of $y$ samples

 **end for**

 Let $A = \{a_{ij}\}$ and $B = \{b_{ij}\}$.

 **(2)** Calculate Multiscale Correlation Map $\mathcal{C}$ & MGC Test Statistic:

 $[c^*, \mathcal{C}, k, l] = \text{MGCSAMPLESTAT}(A, B)$ Algorithm C2

 **(3)** Calculate the p-value

 $pval(c^*) = \text{PERMUTATIONTEST}(A, B, r, c^*)$ Algorithm C3

 **end Function**

---

**Pseudocode C2** MGC test statistic. This algorithm computes all local correlations, take the smoothed maximum, and reports the $(k, l)$ pair that achieves it. For the smoothing step, it: (i) finds the largest connected region in the correlation map, such that each correlation is significant, that is larger than a certain threshold to avoid correlation inflation by sample noise, (ii) take the largest correlation in the region, (iii) if the region area is too small, or the smoothed maximum is no larger than the global correlation, the global correlation is used instead. The running time is $\mathcal{O}(n^2)$.

---

**Input:** A pair of distance matrices $(A, B) \in \mathbb{R}^{n \times n} \times \mathbb{R}^{n \times n}$.

---

**Output:** The MGC statistic $c^* \in \mathbb{R}$, all local statistics $\mathcal{C} \in \mathbb{R}^{n \times n}$, and the corresponding local scale $(k, l) \in \mathbb{N} \times \mathbb{N}$.

---

*Continued on next page*

| | | |
|---|---|---|
| 1: | **function** MGCSAMPLESTAT($A, B$) | |
| 2: | $\mathcal{C} = $ MGCALLLOCAL($A, B$) | All local correlations |
| 3: | $\tau = $ THRESHOLDING($\mathcal{C}$) | find a threshold to determine large local correlations |
| 4: | **for** $i, j := 1, \ldots, n$ **do** $r_{ij} \leftarrow \mathbb{I}(c^{ij} > \tau)$ **end for** | identify all scales with large correlation |
| 5: | $\mathcal{R} \leftarrow \{r_{ij} : i, j = 1, \ldots, n\}$ | binary map encoding scales with large correlation |
| 6: | $\mathcal{R} = $ CONNECTED($\mathcal{R}$) | largest connected component of the binary matrix |
| 7: | $c^* \leftarrow \mathcal{C}(n, n)$ | use the global correlation by default |
| 8: | $k \leftarrow n, l \leftarrow n$ | |
| 9: | **if** $\left(\sum_{i,j} r_{ij}\right) \geq 2n$ **then** | proceed when the significant region is sufficiently large |
| 10: | $[c^*, k, l] \leftarrow \max(\mathcal{C} \circ \mathcal{R})$ | find the smoothed maximum and the respective scale |
| 11: | **end if** | |
| 12: | **end Function** | |

**Input:** $\mathcal{C} \in \mathbb{R}^{n \times n}$.

**Output:** A threshold $\tau$ to identify large correlations.

| | | |
|---|---|---|
| 13: | **function** THRESHOLDING $\mathcal{C}$ | |
| 14: | $\tau \leftarrow \sum_{c^{ij} < 0}(c^{ij})^2 / \sum_{c^{ij} < 0} 1$ | variance of all negative local generalized correlations |
| 15: | $\tau \leftarrow \max\{0.01, \sqrt{\tau}\} \times 3.5$ | threshold based on negative correlations |
| 16: | $\tau \leftarrow \max\{\tau, 2/n, c^{nn}\}$ | |
| 17: | **end Function** | |

**Pseudocode C3** Permutation Test. This algorithm uses the random permutation test with $r$ random permutations for the p-value, requiring $\mathcal{O}(m^2 \log n)$ for MGC. In the real-data experiment, we always set $r = 10,000$. Note that the p-value computation for any other global generalized correlation coefficient follows from the same algorithm by replacing MGC with the respective test statistic.

**Input:** A pair of distance matrices $(A, B) \in \mathbb{R}^{n \times n} \times \mathbb{R}^{n \times n}$, the number of permutations $r$, and MGC statistic $c^*$ for the observed data.

**Output:** The p-value $pval \in [0, 1]$.

| | | |
|---|---|---|
| 1: | **function** PERMUTATIONTEST($A, B, r, c^*$) | |
| 2: | **for** $t := 1, \ldots, r$ **do** | |
| 3: | $\pi = $ RANDPERM($n$) | generate a random permutation of size $n$ |
| 4: | $c_0^*[t] = $ MGCSAMPLESTAT($A, B(\pi, \pi)$) | calculate the permuted MGC statistic |
| 5: | **end for** | |
| 6: | $pval(c^*) \leftarrow \frac{1}{t} \sum_{t=1}^{r} \boldsymbol{I}(c^* \leq c_0^*[t])$ | compute p-value of MGC |
| 7: | **end function** | |

**Pseudocode C4** Compute local test statistic at a given scale. This algorithm runs in $\mathcal{O}(n^2)$ once the rank information is provided, which is suitable for MGC computation if an optimal scale is already estimated. But it would take $\mathcal{O}(n^4)$ if used to compute all local generalized correlations. Note that for the default MGC implementation uses single centering, the centering function centers $A$ by column and $B$ by row, and the sorting function sorts $A$ within column and $B$ within row. By utilizing $T = \log(n)$ cores, the sorting function can be easily parallelized to take $\mathcal{O}(n^2 \log(n)/T) = \mathcal{O}(n^2)$.

**Input:** A pair of distance matrices $(A, B) \in \mathbb{R}^{n \times n} \times \mathbb{R}^{n \times n}$, and a local scale $(k, l) \in \mathbb{N} \times \mathbb{N}$.

**Output:** The local generalized correlation coefficient $c^{kl} \in [-1, 1]$.

| | | |
|---|---|---|
| 1: | **function** LOCALGENCORR($A, B, k, l$) | |
| 2: | **for** $Z := A, B$ **do** $\mathcal{E}^Z = $ SORT($Z$) **end for** | parallelized sorting |
| 3: | **for** $Z := A, B$ **do** $Z = $ CENTER($Z$) **end for** | center distance matrices |
| 4: | $\tilde{c}^{kl} \leftarrow tr((A \circ \mathcal{E}^A)^{\mathrm{T}} \times (B \circ (\mathcal{E}^B)^{\mathrm{T}}))$ | un-normalized local distance covariance |

*Continued on next page*

5: $v^A \leftarrow tr((A \circ \mathcal{E}^A)^{\mathrm{T}} \times (A \circ (\mathcal{E}^A)^{\mathrm{T}}))$ local distance variances

6: $v^B \leftarrow tr((B \circ \mathcal{E}^B)^{\mathrm{T}} \times (B \circ (\mathcal{E}^B)^{\mathrm{T}}))$

7: $e^A \leftarrow \sum_{i,j=1}^{n}(A \circ \mathcal{E}^A)_{ij}$ sample means

8: $e^B \leftarrow \sum_{i,j=1}^{n}(B \circ \mathcal{E}^B)_{ij}$

9: $c^{kl} \leftarrow (\tilde{c}^{kl} - e^A e^B/n^2)/\sqrt{\left(v^A - (e^A/n)^2\right)\left(v^B - (e^B/n)^2\right)}$ center and normalize

10: **end function**

---

**Pseudocode C5** Compute the multiscale correlation map (i.e., all local generalized correlations) in $\mathcal{O}(n^2 \log n/T)$. Once the distances are sorted, the remaining algorithm runs in $\mathcal{O}(n^2)$. An important observation is that each product $a_{ij}b_{ij}$ is included in $c^{kl}$ if and only if $(k,l)$ satisfies $k \leq R(A_{\cdot j}, i)$ and $l \leq R(B_{\cdot j}, i)$, so it suffices to iterate through $a_{ij}b_{ij}$ for $i, j := 1, \ldots, n$, and add the product simultaneously to all $c^{kl}$ whose scales are no more than $(R(A_{\cdot j}, i), R(B_{\cdot j}, i))$. To achieve the above, we iterate through each product, add it to $c^{kl}$ at $(kl) = (R(A_{\cdot j}, i), R(B_{\cdot j}, i))$ only (so only one local scale is accessed for each operation); then add up adjacent $c^{kl}$ for $k, l = 1, \ldots, n$. The same applies to all local covariances, variances, and expectations.

---

**Input:** A pair of distance matrices $(A, B) \in \mathbb{R}^{n \times n} \times \mathbb{R}^{n \times n}$.

---

**Output:** The multiscale correlation map $\mathcal{C} \in [-1, 1]^{n \times n}$ for $k, l = 1, \ldots, n$.

---

1: **function** MGCAllLocal(A, B)

2: **for** $Z := A, B$ **do** $\mathcal{E}^Z = \textsc{Sort}(Z)$ **end for**

3: **for** $Z := A, B$ **do** $Z = \textsc{Center}(Z)$ **end for**

4: **for** $i, j := 1, \ldots, n$ **do** iterate through all local scales to calculate each term

5: $k \leftarrow \mathcal{E}_{ij}^Z$

6: $l \leftarrow \mathcal{E}_{ij}^Z$

7: $\tilde{c}^{kl} \leftarrow \tilde{c}^{kl} + a_{ij}b_{ij}$

8: $v_k^A \leftarrow v_k^A + a_{ij}^2$

9: $v_l^B \leftarrow v_l^B + b_{ij}^2$

10: $e_k^A \leftarrow e_k^A + a_{ij}$

11: $e_l^B \leftarrow e_l^B + b_{ij}$

12: **end for**

13: **for** $k := 1, \ldots, n - 1$ **do** iterate through each scale again and add up adjacent terms

14: $\tilde{c}^{1,k+1} \leftarrow \tilde{c}^{1,k} + \tilde{c}^{1,k+1}$

15: $\tilde{c}^{k+1,1} \leftarrow \tilde{c}^{k+1,1} + \tilde{c}^{k+1,1}$

16: **for** $Z := A, B$ **do** $v_{k+1}^Z \leftarrow v_k^Z + v_{k+1}^Z$ **end for**

17: **for** $Z := A, B$ **do** $e_{k+1}^Z \leftarrow e_k^Z + e_{k+1}^Z$ **end for**

18: **end for**

19: **for** $k, l := 1, \ldots, n - 1$ **do**

20: $\tilde{c}^{k+1,l+1} \leftarrow \tilde{c}^{k+1,l} + \tilde{c}^{k,l+1} + \tilde{c}^{k+1,l+1} - \tilde{c}^{k,l}$

21: **end for**

22: **for** $k, l := 1, \ldots, n$ **do**

23: $c^{kl} \leftarrow (\tilde{c}^{kl} - e_k^A e_l^B/n^2)/\sqrt{\left(v_k^A - e_k^{A2}/n^2\right)\left(v_l^B - e_l^{B2}/n^2\right)}$

24: **end for**

*Continued on next page*

25:        end function

---

**Pseudocode C6** Power computation of MGC against a given distribution. By repeatedly sampling from the joint distribution $F_{XY}$, sample data of size $n$ under the null and the alternative are generated for $r$ Monte-Carlo replicates. The power of MGC follows by computing the test statistic under the null and the alternative using Algorithm C2. In the simulations we use $r = 10{,}000$ MC replicates. Note that power computation for other benchmarks follows from the same algorithm by plugging in the respective test statistic.

---

**Input:** A joint distribution $F_{XY}$, the sample size $n$, the number of MC replicates $r$, and the type 1 error level $\alpha$.

**Output:** The power $\beta$ of MGC.

---

1:    function MGCPOWER($F_{XY}, n, r, \alpha$)

2:        for $t := 1, \ldots, r$ do

3:            for $i := [n]$ do

4:                $x_i^0 \overset{iid}{\sim} F_X, \; y_i^0 \overset{iid}{\sim} F_Y$                                            sample from null

5:                $(x_i^1, y_i^1) \overset{iid}{\sim} F_{XY},$                                            sample from alternative

6:            end for

7:            for $i, j := 1, \ldots, n$ do

8:                $a_{ij}^0 = \delta_x(x_i^0, x_j^0), \; b_{ij}^0 = \delta_y(y_i^0, y_j^0)$                    pairwise distances under the null

9:                $a_{ij}^1 = \delta_x(x_i^1, x_j^1), \; b_{ij}^1 = \delta_y(y_i^1, y_j^1)$                    pairwise distances under the alternative

10:           end for

11:           $c_0^*[t] = \text{MGCSAMPLESTAT}(A^0, B^0)$                                   MGC statistic under the null

12:           $c_1^*[t] = \text{MGCSAMPLESTAT}(A^1, B^1)$                                   MGC statistic under the alternative

13:        end for

14:        $\omega_\alpha \leftarrow \text{CDF}_{1-\alpha}(c_0^*[t], t \in [r])$                         the critical value of MGC under the null

15:        $\beta \leftarrow \sum_{t=1}^r (c_1^*[t] > \omega_\alpha)/r$                      compute power by the alternative distribution

16:    end function

---

## Simulation dependence functions

This section provides the 20 different dependency functions used in the simulations. We used essentially the exact same relationships as previous publications to ensure a fair comparison (*Székely et al., 2007*; *Simon and Tibshirani, 2012*; *Gorfine et al., 2012*). We only made changes to add white noise and a weight vector for higher dimensions, thereby making them more difficult, to better compare all methods throughout different dimensions and sample sizes. A few additional relationships are also included.

For each sample $x \in \mathbb{R}^p$, we denote $x_{[d]}, d = 1, \ldots, p$ as the $d^{th}$ dimension of the vector $x$. For the purpose of high-dimensional simulations, $w \in \mathbb{R}^p$ is a decaying vector with $w_{[d]} = 1/d$ for each $d$, such that $w^T x$ is a weighted summation of all dimensions of $x$. Furthermore, $\mathcal{U}(a, b)$ denotes the uniform distribution on the interval $(a, b)$, $\mathcal{B}(p)$ denotes the Bernoulli distribution with probability $p$, $\mathcal{N}(\mu, \Sigma)$ denotes the normal distribution with mean $\mu$ and covariance $\Sigma$, $U$ and $V$ represent some auxiliary random variables, $\kappa$ is a scalar constant to control the noise level (which equals 1 for one-dimensional simulations and 0 otherwise), and $\epsilon$ is a white noise from independent standard normal distribution unless mentioned otherwise.

For all the below equations, $(X, Y) \overset{iid}{\sim} F_{XY} = F_{Y|X} F_X$. For each relationship, we provide the space of $(X, Y)$, and define $F_{Y|X}$ and $F_X$, as well as any additional auxiliary distributions.

1. Linear $(X, Y) \in \mathbb{R}^p \times \mathbb{R}$,

$$X \sim \mathcal{U}(-1, 1)^p,$$
$$Y = w^T X + \kappa \epsilon.$$

2. Exponential $(X, Y) \in \mathbb{R}^p \times \mathbb{R}$:

$$X \sim \mathcal{U}(0,3)^p,$$
$$Y = exp(w^T X) + 10\kappa\epsilon.$$

3. Cubic $(X,Y) \in \mathbb{R}^p \times \mathbb{R}$:

$$X \sim \mathcal{U}(-1,1)^p,$$
$$Y = 128(w^T X - \frac{1}{3})^3 + 48(w^T X - \frac{1}{3})^2 - 12(w^T X - \frac{1}{3}) + 80\kappa\epsilon.$$

4. Joint normal $(X,Y) \in \mathbb{R}^p \times \mathbb{R}^p$: Let $\rho = 1/2p$, $I_p$ be the identity matrix of size $p \times p$, $J_p$ be the matrix of ones of size $p \times p$, and $\Sigma = \begin{bmatrix} I_p & \rho J_p \\ \rho J_p & (1 + 0.5\kappa)I_p \end{bmatrix}$. Then

$$(X,Y) \sim \mathcal{N}(0, \Sigma).$$

5. Step Function $(X,Y) \in \mathbb{R}^p \times \mathbb{R}$

$$X \sim \mathcal{U}(-1,1)^p,$$
$$Y = \boldsymbol{I}(w^T X > 0) + \epsilon,$$

where $\boldsymbol{I}$ is the indicator function, that is $\boldsymbol{I}(z)$ is unity whenever $z$ true, and zero otherwise.

6. Quadratic $(X,Y) \in \mathbb{R}^p \times \mathbb{R}$:

$$X \sim \mathcal{U}(-1,1)^p,$$
$$Y = (w^T X)^2 + 0.5\kappa\epsilon.$$

7. W Shape $(X,Y) \in \mathbb{R}^p \times \mathbb{R} : U \sim \mathcal{U}(-1,1)^p,$

$$X \sim \mathcal{U}(-1,1)^p,$$
$$Y = 4\left[ \left( (w^T X)^2 - \frac{1}{2} \right)^2 + w^T U / 500 \right] + 0.5\kappa\epsilon.$$

8. Spiral $(X,Y) \in \mathbb{R}^p \times \mathbb{R} : U \sim \mathcal{U}(0,5), \epsilon \sim \mathcal{N}(0,1)$

$$X_{[d]} = U \sin(\pi U) \cos^d(\pi U) \text{ for } d = 1, \ldots, p-1,$$
$$X_{[d]} = U \cos^p(\pi U),$$
$$Y = U \sin(\pi U) + 0.4 p\epsilon.$$

9. Uncorrelated Bernoulli $(X,Y) \in \mathbb{R}^p \times \mathbb{R} : U \sim \mathcal{B}(0.5) \, \epsilon_1 \sim \mathcal{N}(0,I_p), \epsilon_2 \sim \mathcal{N}(0,1),$

$$X \sim \mathcal{B}(0.5)^p + 0.5\epsilon_1,$$
$$Y = (2U - 1)w^T X + 0.5\epsilon_2.$$

10. Logarithmic $(X,Y) \in \mathbb{R}^p \times \mathbb{R}^p : \epsilon \sim \mathcal{N}(0,I_p)$

$$X \sim \mathcal{N}(0,I_p),$$
$$Y_{[d]} = 2\log_2(|X_{[d]}|) + 3\kappa\epsilon_{[d]} \text{ for } d = 1, \ldots, p.$$

11. Fourth Root $(X,Y) \in \mathbb{R}^p \times \mathbb{R}^p$:

$$X \sim \mathcal{U}(-1,1)^p,$$
$$Y = |w^T X|^{\frac{1}{4}} + \frac{\kappa}{4}\epsilon.$$

12. Sine Period $4\pi (X,Y) \in \mathbb{R}^p \times \mathbb{R}^p : U \sim \mathcal{U}(-1,1), V \sim \mathcal{N}(0,1)^p, \theta = 4\pi,$

$$X_{[d]} = U + 0.02 p V_{[d]} \text{ for } d = 1, \ldots, p,$$
$$Y = \sin(\theta X) + \kappa\epsilon.$$

13. Sine Period $16\pi (X,Y) \in \mathbb{R}^p \times \mathbb{R}^p$: Same as above except $\theta = 16\pi$ and the noise on $Y$ is changed to $0.5\kappa\epsilon$.

14. Square $(X,Y) \in \mathbb{R}^p \times \mathbb{R}^p$: Let $U \sim \mathcal{U}(-1,1)$, $V \sim \mathcal{U}(-1,1)$, $\epsilon \sim \mathcal{N}(0,1)^p$, $\theta = -\frac{\pi}{8}$. Then

$$X_{[d]} = U\cos\theta + V\sin\theta + 0.05p\epsilon_{[d]},$$
$$Y_{[d]} = -U\sin\theta + V\cos\theta,$$

**for** $d = 1, \ldots, p$.

15. Two Parabolas $(X, Y) \in \mathbb{R}^p \times \mathbb{R}$: $\epsilon \sim \mathcal{U}(0, 1)$, $U \sim \mathcal{B}(0.5)$,

$$X \sim \mathcal{U}(-1, 1)^p,$$
$$Y = \left( (w^{\mathrm{T}} X)^2 + 2\kappa\epsilon \right) \cdot \left( U - \frac{1}{2} \right).$$

16. Circle $(X, Y) \in \mathbb{R}^p \times \mathbb{R}$: $U \sim \mathcal{U}(-1, 1)^p$, $\epsilon \sim \mathcal{N}(0, I_p)$, $r = 1$,

$$X_{[d]} = r\left( \sin(\pi U_{[d+1]}) \prod_{j=1}^{d} \cos(\pi U_{[j]}) + 0.4\epsilon_{[d]} \right) \textbf{ for } d = 1, \ldots, p-1,$$

$$X_{[p]} = r\left( \prod_{j=1}^{p} \cos(\pi U_{[j]}) + 0.4\epsilon_{[p]} \right),$$

$$Y = \sin(\pi U_{[1]}).$$

17. Ellipse $(X, Y) \in \mathbb{R}^p \times \mathbb{R}$: Same as above except $r = 5$.
18. Diamond $(X, Y) \in \mathbb{R}^p \times \mathbb{R}^p$: Same as 'Square' except $\theta = -\frac{\pi}{4}$.
19. Multiplicative Noise $(x, y) \in \mathbb{R}^p \times \mathbb{R}^p$: $u \sim \mathcal{N}(0, I_p)$,

$$x \sim \mathcal{N}(0, I_p),$$
$$y_{[d]} = u_{[d]} x_{[d]} \textbf{ for } d = 1, \ldots, p.$$

20. Multimodal Independence $(X, Y) \in \mathbb{R}^p \times \mathbb{R}^p$: Let $U \sim \mathcal{N}(0, I_p)$, $V \sim \mathcal{N}(0, I_p)$, $U' \sim \mathcal{B}(0.5)^p$, $V' \sim \mathcal{B}(0.5)^p$. Then

$$X = U/3 + 2U' - 1,$$
$$Y = V/3 + 2V' - 1.$$

For each distribution, $X$ and $Y$ are dependent except (20); for some relationships (8,14,16-18) they are independent upon conditioning on the respective auxiliary variables, while for others they are 'directly' dependent. A visualization of each dependency with $D = D_y = 1$ is shown in *Figure 2— figure supplement 1*.

For the increasing dimension simulation in the main paper, we always set $\kappa = 0$ and $n = 100$, with $p$ increasing. Note that $q = p$ for types 4, 10, 12, 13, 14, 18, 19, 20,, otherwise $q = 1$. The decaying vector $w$ is utilized for $p > 1$ to make the high-dimensional relationships more difficult (otherwise, additional dimensions only add more signal). For the one-dimensional simulations, we always set $p = q = 1$, $\kappa = 1$ and $n = 100$.

## Acknowledgements

This work was partially supported by the Child Mind Institute Endeavor Scientist Program, the National Science Foundation award DMS-1712947, the National Security Science and Engineering Faculty Fellowship (NSSEFF), the Johns Hopkins University Human Language Technology Center of Excellence (JHU HLT COE), the Defense Advanced Research Projects Agency's (DARPA) SIMPLEX program through SPAWAR contract N66001-15-C-4041, the XDATA program of DARPA administered through Air Force Research Laboratory contract FA8750-12-2-0303, DARPA Lifelong Learning Machines program through contract FA8650-18-2-7834, the Office of Naval Research contract N00014-12-1-0601, the Air Force Office of Scientific Research contract FA9550-14-1-0033. The authors thank Dr. Brett Mensh of Optimize Science for acting as our intellectual consigliere, Julia Kuhl for help with figures, and Dr. Ruth Heller, Dr. Bert Vogelstein, Dr. Don Geman, and Dr. Yakir Reshef for insightful suggestions. And we'd like to thank Satish Palaniappan, Sambit Panda, and Junhao (Bear) Xiong for porting the code to Python.

## Additional information

### Funding

| Funder | Grant reference number | Author |
|---|---|---|
| Child Mind Institute | Endeavor Scientist Program | Joshua T Vogelstein |
| National Science Foundation | | Joshua T Vogelstein |
| Defense Advanced Research Projects Agency | | Joshua T Vogelstein |
| Office of Naval Research | | Joshua T Vogelstein |
| Air Force Office of Scientific Research | | Joshua T Vogelstein |

The funders had no role in study design, data collection and interpretation, or the decision to submit the work for publication.

### Author contributions

Joshua T Vogelstein, Conceptualization, Formal analysis, Supervision, Funding acquisition, Investigation, Visualization, Methodology, Writing—original draft, Project administration, Writing—review and editing; Eric W Bridgeford, Software; Qing Wang, Data curation; Carey E Priebe, Supervision; Mauro Maggioni, Conceptualization; Cencheng Shen, Conceptualization, Formal analysis, Investigation, Visualization, Methodology, Writing—review and editing

### Author ORCIDs

Joshua T Vogelstein (iD) http://orcid.org/0000-0003-2487-6237
Cencheng Shen (iD) http://orcid.org/0000-0003-1030-1432

### Decision letter and Author response

Decision letter https://doi.org/10.7554/eLife.41690.029
Author response https://doi.org/10.7554/eLife.41690.030

## Additional files

### Supplementary files

• Transparent reporting form
DOI: https://doi.org/10.7554/eLife.41690.018

### Data availability

To facilitate reproducibility, we make all datasets available from: https://github.com/neurodata/MGC-paper/tree/master/Data/Preprocessed (copy archived at https://github.com/elifesciences-publications/MGC-paper).

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

## Appendix 1

DOI: https://doi.org/10.7554/eLife.41690.019

## Real data processing

### Brain activity vs personality

This experiment investigates whether there is any dependency between resting brain activity and personality. Human personality has been intensively studied for many decades; the most widely used and studied approach is the NEO Personality Inventory-Revised the characterized personality along five dimensions (*Costa and McCrae, 1992*).

This dataset consists of 42 subjects, each with 197 time-steps of resting-state functional magnetic resonance activity (rs-fMRI) activity, as well as the subject's five-dimensional 'personality'. Adelstein et al. (*Adelstein et al., 2011*) were able to detect dependence between the activity of certain brain regions and dimensions of personality, but lacked the tools to test for dependence of whole brain activity against all five dimensions of personality.

For the five-factor personality modality, we used the Euclidean distance. For the brain activity modality, we derived the following comparison function. For each scan, (i) run Configurable Pipeline for the Analysis of Connectomes pipeline (*Craddock et al., 2013*) to process the raw brain images yielding a parcellation into 197 regions of interest, (ii) run a spectral analysis on each region and keep the power of band, (iii) bandpass and normalize it to sum to one, (iv) calculate the Kullback-Leibler divergence across regions to obtain a similarity matrix across comparing all regions. Then, use the normalized Hellinger distance to compute distances between each subject.

### Brain connectivity vs creativity

This experiment investigates whether there is any dependency between brain structural networks and creativity. Creativity has been extensively studied in psychology; the 'creativity composite index' (CCI) is an index similar to an 'intelligence quotient' but for creativity rather than intelligence (*Jung et al., 2009*).

This dataset consists of 109 subjects, each with diffusion weighted MRI data as well as the subject's CCI. Neural correlates of CCI have previously been investigated, though largely using structural MRI and cortical thickness (*Jung et al., 2009*). Previously published results explored the relationship between graphs and CCI (*Koutra et al., 2015*), but did not provide a valid test.

We used Euclidean distance to compare CCI values. For the raw brain imaging data, we derived the following comparison function. For each scan we estimated brain networks from diffusion and structural MRI data via Migraine, a pipeline for estimating brain networks from diffusion data (*Roncal et al., 2013*). We compute the distance between the graphs using the semi-parametric graph test statistic (*Sussman et al., 2012*; *Shen et al., 2017*; *Tang et al., 2017*), embedding each graph into two dimensions and aligning the embeddings via a Procrustes analysis.

### Proteins vs cancer

This experiment investigated whether there is any dependency between abundance levels of peptides in human plasma and the presence of cancers. Selected Reaction Monitoring (SRM) is a targeted quantitative proteomics technique for measuring protein and peptide abundance in complicated biological samples (*Wang et al., 2011*). In a previous study, we used SRM to identify 318 peptides from 33 normal, 10 pancreatic cancer, 28 colorectal cancer, and 24 ovarian cancer samples (*Wang et al., 2017*). Then, using other methods, we identifed three peptides that were implicated in ovarian cancer, and validated them as legitimate biomarkers with a follow-up experiment.

In this study, we performed the following five sets of tests on those data:

1. Ovarian vs. normal for all proteins,
2. Ovarian vs. normal for each individual protein,
3. Pancreas vs. normal for all proteins,
4. Pancreas vs. all others for each individual protein,
5. Pancreas vs. normal for each individual protein.

These tests are designed to first validate the MGC method from ovarian cancer, then identify biomarkers unique to pancreatic cancer, that is, find a protein that is able to tell the difference between pancreas and normals, as well as pancreas vs all other cancers. For each of the five tests, we create a binary label vector, with 1 indicating the cancer type of interest for the corresponding subject, and 0 otherwise. Then each algorithm is applied to each task. For all tests we used Euclidean distances and the type 1 error level is set to $\alpha = 0.05$. The three test sets assessing individual proteins provide 318 p-values; we used the Benjamini-Hochberg procedure (*Benjamini and Hochberg, 1995*) to control the false discovery rate. A summary of the results are reported in *Appendix 1—table 1*.

**Appendix 1—table 1.** Results for cancer peptide screening. The first two rows report the p-values for the tests of interest based on all peptides. The next four rows report the number of significant proteins from individual peptide tests; the Benjamini-Hochberg procedure is used to locate the significant peptides by controlling the false discovery rate at 0.05.

| | Testing pairs / Methods | MGC | MANTEL | DCORR | MCORR | HHG |
|---|---|---|---|---|---|---|
| 1 | Ovar vs. Norm: p-value | 0.0001 | 0.0001 | 0.0001 | 0.0001 | 0.0001 |
| 2 | Ovar vs. Norm: # peptides | 218 | 190 | 186 | 178 | 225 |
| 3 | Pancr vs. Norm: p-value | 0.0082 | 0.0685 | 0.0669 | 0.0192 | 0.0328 |
| 4 | Panc vs. Norm: # peptides | 9 | 7 | 6 | 7 | 11 |
| 5 | Panc vs. All: # peptides | 1 | 0 | 0 | 0 | 3 |
| 6 | # peptides unique to Panc | 1 | 0 | 0 | 0 | 2 |
| 7 | # false positives for Panc | 0 | n/a | n/a | n/a | 1 |

DOI: https://doi.org/10.7554/eLife.41690.020

The following source data is available for Appendix 1—table 2:
**Appendix 1—table 1—Source data 1.** Ovarian testing results.
DOI: https://doi.org/10.7554/eLife.41690.021
**Appendix 1—table 1—Source data 2** Pancreatic testing results.
DOI: https://doi.org/10.7554/eLife.41690.022
**Appendix 1—table 1—Source data 3.** Peptide screening results for pancreatic.
DOI: https://doi.org/10.7554/eLife.41690.023

All methods are able to successfully detect a dependence between peptide abundances in ovarian cancer samples versus normal samples (*Appendix 1—table 1*, line 1). This is likely because there are so many individual peptides that have different abundance distributions between ovarian and normal samples (*Appendix 1—table 1*, line 2). Nonetheless, MGC identified more putative biomarkers than any of the other methods. While we have not checked all of them with subsequent experiments to identify potential false positives, we do know from previous experiments that three peptides in particular are effective biomarkers.

All three peptides have p-value $\approx 0$ for all methods including MGC, that is, they are all correctly identified as significant. However, by ranking the peptides based on the actual test statistic of each peptide, MGC is the method that ranks the three known biomarkers the lowest, suggesting that it is the least likely to falsely identify peptides.

We then investigated the pancreatic samples in an effort to identify biomarkers that are unique to pancreas. We first checked whether the methods could identify a difference using all the peptides. Indeed, three methods found a dependence at the 0.05 level, with MGC obtaining the lowest p-value (*Appendix 1—table 1*, line 3). We then investigated how many individual peptides the methods identified; all of them found 6 to 11 peptides with a

significant difference between pancreatic and normal samples (*Appendix 1—table 1*, line 4). Because we were interested in identifying peptides that were uniquely useful for pancreatic cancer, we then compared pancreatic samples to all others. At significance level 0.05, only Mɢᴄ, Hsɪᴄ, and Hʜɢ identified peptides that expressed different abundances in this more challenging case, and we list the top four peptides in *Appendix 1—table 2* along with the corrected p-value for each peptide.

**Appendix 1—table 2.** For each of Mɢᴄ, Dᴄᴏʀʀ, Mᴄᴏʀʀ, Hʜɢ, Hsɪᴄ, Mᴀɴᴛᴇʟ, Pᴇᴀʀsᴏɴ, and Mɪᴄ, list the top four peptides identified for Panc vs All and the respective corrected p-value using Benjamini-Hochberg. Bold indicates a significant peptide at type 1 error level 0.05. The top candidates are very much alike except Mɪᴄ. In particular, neurogranin is consistently among the top candidates for all methods, but is only significant while using Mɢᴄ, Hsɪᴄ, and Hʜɢ; there are two other significant proteins from Hsɪᴄ and Hʜɢ, but they do not further improve the classification performance comparing to just using neurogranin. Note that the p-values from Mᴀɴᴛᴇʟ and Pᴇᴀʀsᴏɴ are always 1 after Benjamini-Hochberg correction, so their respective top peptides are identified using raw p-values without correction.

| method | Top four identified peptides | | | |
|---|---|---|---|---|
| **Mɢᴄ** | neurogranin | fibrinogen protein 1 | tropomyosin alpha-3 | ras suppressor protein 1 |
| **p-value** | 0.03 | 0.33 | 0.49 | 0.52 |
| **Dᴄᴏʀʀ** | neurogranin | fibrinogen protein 1 | kinase 6 | twinfilin-2 |
| **p-value** | 0.41 | 0.60 | 0.60 | 0.93 |
| **Mᴄᴏʀʀ** | neurogranin | fibrinogen protein 1 | kinase 6 | tropomyosin alpha-3 |
| **p-value** | 0.45 | 0.80 | 0.80 | 0.83 |
| **Hsɪᴄ** | neurogranin | tropomyosin alpha-3 | kinase 6 | tripeptidyl-peptidase 2 |
| **p-value** | 0.01 | 0.01 | 0.09 | 0.09 |
| **Hʜɢ** | neurogranin | fibrinogen protein 1 | tropomyosin alpha-3 | platelet basic protein |
| **p-value** | 0.03 | 0.03 | 0.03 | 0.11 |
| **Mᴀɴᴛᴇʟ** | neurogranin | adenylyl cyclase | tropomyosin alpha-3 | alpha-actinin-1 |
| **p-value** | 1 | 1 | 1 | 1 |
| **Pᴇᴀʀsᴏɴ** | neurogranin | adenylyl cyclase | tropomyosin alpha-3 | alpha-actinin-1 |
| **p-value** | 1 | 1 | 1 | 1 |
| **Mɪᴄ** | kinase B | S100-A9 | ERF3A | thymidine |
| **p-value** | 0.15 | 0.15 | 0.15 | 0.15 |

DOI: https://doi.org/10.7554/eLife.41690.024

All three methods reveal the same unique protein for pancreas: neurogranin. Hsɪᴄ identifies another peptide (tropomyosin alpha-3 chain isoform 4), and Hʜɢ identifies a third peptide (fibrinogen-like protein 1 precursor). However, fibrinogen-like protein 1 precursor is not significant for p-value testing between pancreatic and normal subjects. On the other hand, tropomyosin is a ubiquitously expressed protein, since normal tissues and other cancers will also express tropomyosin and leak it into blood, whereas neurogranin is exclusively expressed only in brain tissues. Moreover, there exists strong evidence of tropomyosin 3 upregulated in other cancers (*Karsani et al., 2014*; *Sun et al., 2016*; *Lee et al., 2012*; *Lam et al., 2012*). Therefore, it suggests that the other two peptides identified by Hʜɢ and Hsɪᴄ are likely false positives.

In fact, neurogranin is always one of the top 4 candidates in all methods except Mɪᴄ; the only difference is that the corrected p-values are not significant enough for other methods. Along with the classification result in *Figure 4D* showing that neurogranin alone has the best classification error, Mɢᴄ discovers an ideal candidate for potential biomarker. Moreover, the fact that Mɢᴄ, Hʜɢ and Hsɪᴄ discover the dependency while others cannot implies a nonlinear relationship.

## MGC does not inflate false positive rates in screening

In this final experiment, we empirically determine that MGC does not inflate false positive rates via a neuroimaging screening. To do so, we extend the work of Eklund et al. (*Eklund et al., 2012*; *Eklund et al., 2016*), where a number of parametric methods are shown to largely inflate the false positives. Specifically, we applied MGC to test whether there is any dependency between brain voxel activities and random numbers. For each brain region, MGC attempts to test the following hypothesis: Is activity of a brain region independent of the time-varying stimuli? Any region that is selected as significant is a false positive by construction. By testing each brain region separately, MGC provides a distribution of false positive rates. If MGC is valid, the resulting distribution should be centered around the significance level, which is set at 0.05 for these experiments.

We considered 25 resting state fMRI experiments from the 1000 Functional Connectomes Project consisting of a total of 1583 subjects (*Biswal et al., 2010*). *Appendix 1—figure 1* shows the false positive rates of MGC for each dataset, which are centered around the critical level 0.05, as it should be. In contrast, many standard parametric methods for fMRI analysis, such as generalized linear models, can significantly increase the false positive rates, depending on the data and pre-processing details (*Eklund et al., 2012*; *Eklund et al., 2016*). Moreover, even the proposed solutions to those issues make linearity assumptions, thereby limiting detection to only a small subset of possible dependence functions.

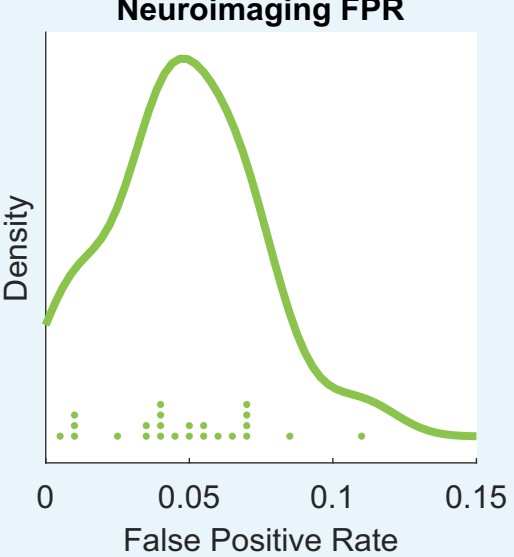

**Appendix 1—figure 1.** We demonstrate that MGC is a valid test that does not inflate the false positives in screening and variable selection. This figure shows the density estimate for the false positive rates of applying MGC to select the 'falsely significant' brain regions versus independent noise experiments; dots indicate the false positive rate of each experiment. The mean ± standard deviation is $0.0538 \pm 0.0394$.

DOI: https://doi.org/10.7554/eLife.41690.025

## Running time report in experiments

*Appendix 1—table 3* lists the actual running time of MGC versus other methods for testing on the real data, based on a modern desktop with a six core I7-6850K CPU and 32 GB memory on MATLAB 2017a on Windows 10. The first two experiments are timed based on 1000 permutations, while the screening experiment is timed without permutation, that is compute the test statistic only. Pearson runs the fastest, trailed by MIC and then DCORR. PEARSON and MIC are only possible to run in the screening experiment, as the other two experiments are multivariate. The running time of MGC is a constant times (about 10) higher

than that of D$_{\text{CORR}}$, and H$_{\text{HG}}$ is implemented in a running time of $O(n^3)$ and thus significantly slower.

**Appendix 1—table 3.** The actual testing time (in seconds) on real data.

| Data | Personality | Creativity | Screening |
|---|---|---|---|
| M$_{\text{GC}}$ | 2.5 | 7.5 | 1.9 |
| D$_{\text{CORR}}$ | 0.2 | 0.4 | 0.18 |
| H$_{\text{SIC}}$ | 0.5 | 1.7 | 0.23 |
| H$_{\text{HG}}$ | 6.3 | 53.4 | 12.3 |
| P$_{\text{EARSON}}$ | NA | NA | 0.03 |
| M$_{\text{IC}}$ | NA | NA | 0.1 |
| M$_{\text{RULE}}$ | | | |

DOI: https://doi.org/10.7554/eLife.41690.026

