## [Decision Letter]

Thank you for submitting your article "Discovering and Deciphering Relationships Across Disparate Data Modalities" for consideration by *eLife*. Your article has been reviewed by two peer reviewers, and the evaluation has been overseen by a Reviewing Editor, Dane Taylor, and a Senior Editor, Joshua Gold. The following individual involved in the review of your submission has agreed to reveal their identity: James D Wilson (Reviewer #2).

The Reviewers and Reviewing Editor have discussed the reviews with one another and generally found the paper to be well written, clearly organized, and an important advancement to the data-analytics community. While we do not accept the article for publication in its current form, we invite you to resubmit the manuscript after taking the following issues into consideration.

Summary:

In this work, the authors Vogelstein et al. introduce a new technique called Multiscale Graph Correlation (MGC) to discover (i.e., statistically infer using locally biased, distance-based hypothesis testing) and decipher (e.g., study the geometry of) pairwise relationships between different modalities of a dataset. Their approach is straightforward, principled, and computationally reasonable. It improves upon widely used relationship-inference techniques including test statistics for detecting linear relationships (e.g., Pearson's correlation coefficient) and nonlinear relationships (e.g., distance-correlation-based analyses including kNN-based methods, kernel-based methods and Mantel's test). Specifically, it provides an improved null-hypothesis relationship-test statistic that requires fewer samples and can be implemented at a marginal increase in computational cost [the algorithm scales as O(n^2^log *n*) for n data points. The authors study the MGC approach using a synthetic dataset comprised on 20 models, finding MGC to typically outperform competing approaches. In addition to identifying pairwise relationships between datasets, the MGC approach is a multiscale analysis and identifies the spatial scale at which the relationship's inference is most powerful. In doing so, the approach yields an MGC-map that provides rich insight into the nature of the relationship (particularly its geometry). To conclude, the authors apply the MGC test statistic to explore relationships for three biological datasets: brain activity and personality, brain connectivity and creativity, and proteomic biomarkers and cancers.

Essential revisions:

1) The experimental section is lacking sufficient discussion and citation to the literature on related scientific studies. For example, there are several statements in paragraph 2 of subsection “MGC Identifies Potential Cancer Proteomics Biomarkers” that need citation. (Discussion and citations for the brain study also appear to be missing.)

2) The authors should provide more detail about how this method scales up to larger datasets. It should be noted that the provided examples are restricted to small datasets, n\le1000. What are the practical limitations on how you might adapt your algorithm to larger data? It would be helpful to provide further results on computational time, similar to Table 4 in the Appendix. In particular, can the authors provide numerical support for their O(*n*^2^log *n*) scaling result.

3) The proteomics study and discussion is lacking exploration. For example: Were any biomarkers besides neurogranin identified as significant and biologically relevant? What are some of the top hits in Panc vs All and Panc VS norm individually? Also, what are some of the top hits for both of these comparisons? Do these top hits also make sense biologically? Also, you mention that "the rest of the global methods did not identify any markers." Even if no markers were identified to be statistically significant with other methods, could you still consider their relative ranking? In particular, does neurogranin appear in other methods as one of the more important biomarkers? What are the other additional top hits identified by other methods? Where do those hits appear on the scatter plot in Figure 4C?

4) Step 3 of the MGC algorithm (see subsection “The Multiscale Graph Correlation Procedure”) could use further discussion/motivation. First, the local generalized correlation is normalized in the Appendix but not in the subsection “The Multiscale Graph Correlation Procedure”, which is confusing. Second, the local generalized distance correlations are computed using the intersection of the two graphs (the k-nn and the l-nn graphs). That is, the product of terms *A(i,j) G_k_(i,j) B(i,j) H_l_(i,j)* is nonzero if *(i,j)* are in the "closest" neighborhood for both the graph associated with adjacency matrix G_k_ as well as that associated with *H_l_*. Could this be too stringent? That is, couldn't there be a significant correlation between two data sets where only one of the data modality coordinates is in the nearest neighbors? In particular, an entry

A(i,j) G_k_(i,j) B(i,j) H_l_(i,j)

is treated as the same in the following 3 situations:

a) G_k_(i,j) = H_l_(i,j) = 0

b) G_k_(i,j) = 0, H_l_(i,j) = 1

c) G_k_(i,j) = 1, H_l_(i,j) = 0

Shouldn't case (a) be treated differently than cases (b) and (c)? It may be useful for the authors to discuss this and explain their choice.

5) For the original choice of k-nearest neighbors, in practice, how does one choose the distance metric (of course Euclidean is often selected by default)? I know it is context dependent, but is there any general data driven advice for this? The reason why I state this is that one could, if they wanted, basically p-value hunt by choosing the right metric to give a small p-value.

---

## [Author Response]

Essential revisions:1) The experimental section is lacking sufficient discussion and citation to the literature on related scientific studies. For example, there are several statements in paragraph 2 of subsection “MGC Identifies Potential Cancer Proteomics Biomarkers” that need citation. (Discussion and citations for the brain study also appear to be missing.)

For the proteomics study, subsection “MGC Identifies Potential Cancer Proteomics Biomarkers”, we rephrased and added new citations: Bhat et al., 2012, Frantzi et al., 2014, Wang et al., 2011, Willemse et al., 2018, Yang et al., 2015. The complete text is below:

“MGC can also be useful for a completely complementary set of scientific questions: screening proteomics data for biomarkers, often involving the analysis of tens of thousands of proteins, peptides, or transcripts in multiple samples representing a variety of disease types. […] The rest of the global methods did not identify any markers, see Materials and method for more details and Table 4 for identified peptide information using each method.”

We also added a few sentences and citations to the brain imaging study in subsection “MGC Discovers the Relationships between Brain and Mental Properties”:

“However, the relationship between brain activity and structure, and these aspects of our psyche, remains unclear (Deyoung et al., 2010; Xu and Potenza, 2012; Bjørnebekk et al., 2013; Sampaio et al., 2014). For example, prior work did not evaluate the relationship between entire brain connectivity to all five factors of the standard personality model (Costa and McCraw, 1992).”

2) The authors should provide more detail about how this method scales up to larger datasets. It should be noted that the provided examples are restricted to small datasets, n\le1000. What are the practical limitations on how you might adapt your algorithm to larger data? It would be helpful to provide further results on computational time, similar to Table 4 in the Appendix. In particular, can the authors provide numerical support for their O(n^2^log n) scaling result.

Thank you for the valuable points here. Yes, applying the method directly to millions of data can be slow due to the *n*^2^ operation and the permutation test. There are a number of ways to reduce the running time for DCORR and HSIC, which is equally applicable to MGC and we are currently working on incorporating all such fast implementations to MGC. We have now modified the Discussion fourth paragraph:

“Recent advances in related work demonstrated that one could reduce computational time of distance-based tests to close to linear via faster implementation, subsampling, random projection, and null distribution approximation (Huo and Szekely, 2016; Huang and Huo, 2017; Zhang et al., 2017; Chaudhuri and Wenhao, 2018), making it feasible for large amount of data.”

The simulation time comparison can be found in Figure 6 in Shen and Vogelstein, 2018, which validates that MGC has almost the same complexity as DCORR and HSIC and they differ by a constant. For convenience, it is attached here in Author response image 1, and pointed to in subsection “MGC is Computationally Efficient”:

“…in practice, MGC can be *O(n*^2^), meaning only a constant factor slower than DCORR and HSIC, which is illustrated in Figure 6 of Shen et al., 2018.”

**Author response image 1. respfig1:** Compute the test statistics of MGC, DCORR, and HSIC for 100 replicates, and then plot the average running time in log scale (clocked using Matlab 2017a on a Windows 10 machine with I7 six-core CPU). The sample data are repeatedly generated using the quadratic relationship in Appendix, the sample size increases from 25 to 500, and the dimensionality is fixed at *p* = 1 on the left and *p* = 1000 on the right. In either panel, the three lines differ by some constants in the log scale, suggesting the same running time complexity but different constants. MGC has a higher intercept than the other two, which translates to about a constant of 6 times of DCORR and 3 times of HSIC at *n* = 500 and *p* = 1, and about 3 at *p* = 1000.

3) The proteomics study and discussion is lacking exploration. For example: Were any biomarkers besides neurogranin identified as significant and biologically relevant? What are some of the top hits in Panc vs All and Panc VS norm individually? Also, what are some of the top hits for both of these comparisons? Do these top hits also make sense biologically? Also, you mention that "the rest of the global methods did not identify any markers." Even if no markers were identified to be statistically significant with other methods, could you still consider their relative ranking? In particular, does neurogranin appear in other methods as one of the more important biomarkers? What are the other additional top hits identified by other methods? Where do those hits appear on the scatter plot in Figure 4C?

Yes, MGC, HSIC, and HHG all identify neurogranin, while HSIC and HHG identify another two peptides. We do not know the ground truth here, but the other two peptides are related to other cancer types; and a leave-one-out classification also supports using neurogranin alone. The rest of the global methods do not identify any markers after multiple testing corrections, but actually mostly coincide with the MGC discovery in terms of relative ranking. We provide a table (Table 4) for Panc vs All containing the top 4 ranked peptides for each method, (in scatter plot they are the 4 dots with least p-value along the y-axis for MGC method). We first state in paragraph two of subsection “MGC Identifies Potential Cancer Proteomics Biomarkers”:

“In comparison, HSIC identified neurogranin as well, but it also identified another peptide; HHG identified the same two by HSIC, and a third peptide. […] The rest of the global methods did not identify any markers, see Materials and methods for more details and Table 4 for identified peptide information using each method.”

In subsection “Proteins vs Cancer”, we add the table and point out that:

“…Because we were interested in identifying peptides that were uniquely useful for pancreatic cancer, we then compared pancreatic samples to all others. […] Moreover, the fact that MGC, HHG and HSIC discover the dependency while others cannot implies a nonlinear relationship.”

4) Step 3 of the MGC algorithm (see subsection “The Multiscale Graph Correlation Procedure”) could use further discussion/motivation. First, the local generalized correlation is normalized in the Appendix but not in the subsection “The Multiscale Graph Correlation Procedure”, which is confusing.

Thank you for pointing out this discrepancy! We corrected the main text where we define the local correlation measure to the normalized version:

“For all possible values of *k* and *l* from 1 to *n*:

a) Compute the *k*-nearest neighbor graphs for one property, and the *l*-nearest neighbor graphs for the other property. Let *Gk* and *Hl* be the adjacency matrices for the nearest neighbor graphs, so that *Gk (i, j*) = 1 indicates that *A (i, j*) is within the *k* smallest values of the *i^th^*row of *A*, and similarly for *Hl*.

b) Estimate the local correlations—the correlation between distances restricted to only the (*k, l*) neighbors—by summing the products of the above matrices,

ckl=∑ijAi,jGki,jBi,jHl(i,j)

c) Normalize *c^kl^*such that the result is always between *−*1 and +1 by dividing by

∑ijA2(ij)Gk(i,j)x∑ijB2(i,j)Hl(i,j)”

Second, the local generalized distance correlations are computed using the intersection of the two graphs (the k-nn and the l-nn graphs). That is, the product of terms A(i,j) G_k_(i,j) B(i,j) H_l_(i,j) is nonzero if (i,j) are in the "closest" neighborhood for both the graph associated with adjacency matrix G_k_ as well as that associated with H_l_. Could this be too stringent? That is, couldn't there be a significant correlation between two data sets where only one of the data modality coordinates is in the nearest neighbors? In particular, an entryA(i,j) G_k_(i,j) B(i,j) H_l_(i,j)is treated as the same in the following 3 situations:a) G_k_(i,j) = H_l_(i,j) = 0b) G_k_(i,j) = 0, H_l_(i,j) = 1c) G_k_(i,j) = 1, H_l_(i,j) = 0Shouldn't case (a) be treated differently than cases (b) and (c)? It may be useful for the authors to discuss this and explain their choice.

The reason we consider the k-nearest-neighbor and l-nearest-neighbor is that most nonlinear relationships are intrinsically local linear relationship, where only small distances exhibit strong relationship. Geometrically speaking, if there exists a dependency structure where the large distance pairs are highly linearly correlated, then the nearest- neighbors must also be highly linear correlated after centering by the average distance, so the global correlation suffices in this case. It is actually easy to consider only the furthest neighbors in MGC by simply reverting the ranking scheme (so it is sorted in descending order and *k* = 1 includes the largest distance pair), but it does not work better than the global correlation in any simulation, implying the global correlation is able to capture case (a). On the other hand, if small distances in one modality correspond to large distances in another modality, their product after centering is a negative term, causing the test statistic to be smaller. Moreover, since distance correlation is proved larger than 0 if and only if dependency, it cannot happen that small distances always correspond to large distances. We modified the discussion of MGC on in subsection “Local Correlations” to discuss this point.

“As most nonlinear relationships intrinsically exhibit a local linear structure, considering the nearest-neighbors is able to amplify the dependency signal over the global correlation. […] Therefore considering the nearest-neighbor may significantly improve the performance over global correlation, while considering the other scenarios does not.”

5) For the original choice of k-nearest neighbors, in practice, how does one choose the distance metric (of course Euclidean is often selected by default)? I know it is context dependent, but is there any general data driven advice for this? The reason why I state this is that one could, if they wanted, basically p-value hunt by choosing the right metric to give a small p-value.

Thank you for pointing this out. Yes, additional care is needed if one opts to experiment on multiple metric choices. Either one has to correct the smallest p-value for multiple testing, or design a proper procedure to produce a single p-value that correctly controls the type 1 error. We added a paragraph to the Discussion section (second paragraph) to address this issue.

“The default metric choice of MGC in this paper is always the Euclidean distance, other metric choices may be more appropriate in different fields, and using the strong negative type metric as specified in (Lyons, 2013) can guarantee the consistency property. […] Such a testing procedure properly controls the type 1 error level without the need for additional correction.”